



# A 30-meter resolution national urban land-cover dataset of China, 2000–2015

Wenhui Kuang [1], Shu Zhang[1,2], Xiaoyong Li[1,2], Dengsheng Lu[3]

[1]Key Laboratory of Land Surface Pattern and Simulation, Institute of Geographic Sciences and Natural Resources Research,
Chinese Academy of Sciences, Beijing 100101, China.
[2]College of Resources and Environment, University of Chinese Academy of Sciences, Beijing 10049, China.
[3]College of Geographical Sciences, Fujian Normal University, Fuzhou 350117, China.

*Correspondence to*: Wenhui Kuang (kuangwh@igsnrr.ac.cn)

**Abstract.** Accurate urban land-cover datasets are essential for mapping urban environments. However, a series of national
urban land-cover data covering more than 15 years that characterizes urban environments is relatively rare. Here we propose
a hierarchical principle on remotely sensed urban land-use/cover classification for mapping intra-urban structure/component
dynamics. China's Land Use/cover Dataset (CLUD) is updated, delineating the imperviousness, green surface, waterbody
and bare land conditions in cities. A new data subset called CLUD-Urban is created from 2000 to 2015 at five-year intervals
with a medium spatial resolution (30m). The first step is a prerequisite to extract the vector boundaries covered with urban
areas from CLUD. A new method is then proposed using logistic regression between urban impervious surface area (ISA)
and the annual maximum Normalized Difference Vegetation Index (NDVI) value retrieved from Landsat images based on a
big-data platform with Google Earth Engine. National ISA and urban green space (UGS) fraction datasets for China are
generated at 30-meter resolution with five-year intervals from 2000 to 2015. The overall classification accuracy of national
urban areas is 92%. The root mean square error values of ISA and UGS fractions are 0.10 and 0.14, respectively. The
datasets indicate that the total urban area of China was $6.28 \times 10^4\,km^2$ in 2015, with average fractions of 70.70% and 26.54%
for ISA and UGS, respectively. The ISA and UGS increased between 2000 and 2015 with unprecedented annual rates of
1,311.13 $km^2$/yr and 405.30 $km^2$/yr, respectively. CLUD-Urban can be used to enhance our understanding of urbanization
impacts on ecological and regional climatic conditions and urban dwellers' environments. CLUD-Urban can be applied in
future researches on urban environmental research and practices in the future. The datasets can be downloaded from
https://doi.org/10.5281/zenodo.2644932 (Kuang et al., 2019).

## 1 Introduction

The effects of rapid urbanization on environments have been witnessed around the world (Bai et al., 2018) and
profoundly contributed to the changes in biosphere, hydrosphere and atmosphere (Seto et al., 2012; J. Wu et al., 2014;
Kuang et al., 2018). A rapid urbanization process appeared in China in the 21st century which resulted in urban expansion,
impervious surface area (ISA) growth and an impact on urban green space (UGS) (Xu and Min, 2013; Ma et al., 2014; Bai et



al., 2014; Kuang, 2012; Kuang et al., 2013; Kuang et al., 2016). The fast urbanization process further triggered various urban environmental problems, such as urban heat island and urban flooding (Haase et al., 2014; Kuang, 2011; Kuang et al., 2015; Kuang et al., 2017; Zhang et al., 2017). Although more green areas were constructed in Chinese cities recently, China has relatively lower UGS percentage in urban areas than other developed countries (Nowak and Greenfield, 2012; Kuang et al.,

2014). These urban environmental problems triggered the urgency of accurate and high spatial resolution urban land-cover datasets in delineating the underlying urban environments. Along with the development of earth observation technologies, remote sensing technology has become the mainstream method for mapping urban ISA and monitoring its change (Weng, 2012; Wang et al., 2013; Lu et al., 2014).

Various land-use products are freely available worldwide (Grekousis et al., 2015; Dong et al., 2018), such as the

GlobeLand30 (Chen et al., 2015), the University of Maryland (UMD) Land Cover Classification (Hansen et al., 2000), MODIS (Friedl et al., 2010), GlobCover (Bontemps et al., 2011) and FROM-GLC (Gong et al., 2013). These products have different definitions of urban areas or settlements due to their different classification systems, such as The International Geosphere-Biosphere Programme (IGBP) or Food and Agriculture Organization of the United Nations (FAO) (Belward, 1996; FAO, 1997). However, most of them consider urban area to be a homogeneous unit without spatial heterogeneity, i.e.,

no further subdivision for urban area or built-up environment was implemented (Chen et al., 2015).

Detailed ISA inside a city is required as a primary urban environmental index. However, ISA mapping at national scale mainly relies on medium-low resolution remote sensing images, such as AVHRR, MODIS and DMSP-OLS nightlight data (Kuang et al., 2016), with a common spatial resolution of 250–1000 m (Grekousis et al., 2015). For some data, such as GlobeLand30, the resolution can be improved to 30 m with the use of medium-high resolution remote sensing images (Chen

et al., 2015). P. Wang et al. (2017) used a regression tree algorithm to generate a 2010 global urban boundary and internal impervious surface fraction dataset using Landsat images. The U.S. Geological Survey developed the National Land Cover Database (NLCD), which provides ISA fraction, percent tree canopy, land covers and their changes with a spatial resolution of 30 m (Falcone and Homer, 2012). The current publicly available urban land-cover datasets for China integrate either one or two phases of dataset (Gong et al., 2013; Chen et al., 2015) with spatial patterns at coarse resolution (250–1000 m),

making time-series analysis difficult and detail delineation of urban structure impossible. In reality, the remotely sensed urban pixels are mosaicked with ISA, such as buildings, plazas, roads and green land.

The retrieval of ISA and UGS fraction data by unmixing pixels is a pivotal step in mapping urban environments (Kuang et al., 2018). Linear spectral mixture analysis (LSMA) provides an effective method to retrieve the ISA fraction from remote sensing images (Lu and Weng, 2004, 2006). This method has been widely adopted in urban research such as urban internal

structure classification and fraction of urban land cover by Landsat and other medium-high resolution remote sensing images (Peng et al., 2016). However, this method was mainly used for the analysis of a single city or a single urban agglomeration (Zhang and Weng, 2016; Xu et al., 2018). For a larger spatial range, this method becomes time-consuming. Meanwhile, the selection of endmembers relies on manual operation. The knowledge and experience of researchers might influence the selection results to a certain extent, and it is difficult to maintain consistency among different researchers.





In this study, CLUD-Urban is the first dataset giving the possibility for delineating national urban land-cover dynamics in China at high spatial resolution. This dataset provides the urban internal structure and components, including impervious surface area, urban greening space, urban waterbody and bare land at five-year intervals between 2000 and 2015. The retrieval method of urban ISA and UGS fractions was established based on Landsat images providing spatiotemporal patterns.

## 2 Urban land-use classification

### 2.1 Data sources

Medium-high resolution Landsat images were selected as the data source. Landsat is the longest-running satellite series for Earth land observation, from the 1970s, with a total of 8 satellites launched. Except for the early Landsat MSS sensors, Landsat satellites have multispectral bands of 30 m resolution and a continuity in band setting, which is suitable for 1:100,000 mapping requirements. We included Landsat TM, ETM+ and OLI data for analysis and processing. At least 375 images, ranging from paths 118–149 and rows 23–43, were selected for data analysis in each year (Table 1).

We extended dates of images to an interval of three years in order to have more cloud-free images (Table 1). The images from adjacent years were used to supplement the data. Since available image data around 2010 were relatively low, images with similar spectral and spatial resolutions were selected as supplements for data analysis. In our study, the China Brazil Earth Resources Satellite (CBERS-1) and Huan Jing (HJ-1) satellite were selected for 2010.

### 2.2 Urban land extraction

The Chinese Academy of Sciences has updated CLUD at the national level every five years since 2000 (Liu, Liu, Tian et al., 2005; Liu, Liu, Zhuang et al., 2005; Liu et al., 2010), which formed a time series of land-use/cover products at a spatial resolution of 30 m. This land product provides six first-level classes, i.e., cropland, woodland, grassland, water body, built-up area and unused land. The built-up area was divided into three second-level classes: urban land, rural area and industrial and mining land beyond cities. We extracted the vector boundaries of urban land in CLUD's dataset and considered the area within the boundaries to be urban area (Kuang et al., 2016). In this classification system, urban area refers to large, medium and small cities and construction land located in counties and towns. However, this dataset regards a city as a homogeneous unit and cannot reflect intra land-cover status, i.e. ISA, UGS, waterbody and bare land.

The underlying urban condition shows highly spatial heterogeneity (Kuang et al., 2017), which is mosaicked with ISA, UGS, waterbody and bare land. Based on that, we propose the principle of hierarchical structure for urban classification, which can subdivide urban land-cover types into those four categories. Among them, ISA refers to the impervious surface features caused by the impact of artificial land-use activities, like house roofs, asphalt cement roads and parking lots. UGS is an important component of the green infrastructure of cities and provides a range of ecosystem services, as well as cultural services such as recreation and restoration, including parks, trees and grass. Urban waterbody refers to lakes, ponds and





water flows within the city (Kuang et al., 2016). Urban bare land indicates land not covered by vegetation, water, buildings, or roads in cities (Li et al., 2016). UGS and waterbody provide significant, good influences on urban environments (Hamdi and Schayes, 2008), but most urban classification products tend to exclude these two components.

First, we extracted urban areas according to the boundaries established in CLUD. In medium-high resolution images,
urban areas are reflected by the composite characteristics of ISA and UGS, and the spectral characteristics of different urban land uses are not consistent (Fig. 1). For example, the image characteristics of Suzhou old city and new city are different in Landsat 8 OLI images. Because buildings in the old area of the city are distributed compactly, their color in remote sensing images is relatively dark. The new area of the city is dominated by industrial land and was planned well. The city is relatively sprawled out and its features are distinct in the images. With prior knowledge of image classification and human-
computer visual interpretation, we extracted urban land in Suzhou by detecting the city's boundaries.

## 3 Retrieval of ISA and UGS fractions in cities

### 3.1 Pixel-based relationship between ISA and NDVI

A negative correlation between normalized difference vegetation index (NDVI) and ISA fraction was found at the pixel level (Kuang et al., 2016), and this relationship has since been applied in remotely sensed ISA estimation. In arid
regions, however, percentage of vegetation cover is seasonally dependent (Lu et al., 2008). If NDVI is calculated by using only one image, some UGS might be interpreted as bare soil. The fusion of NDVI data in one year may help to improve the accuracy of vegetation characterization. Therefore, multi-interval NDVI data were employed to generate an annual NDVI maximum value. The ISA fraction was estimated through the corresponding empirical model.

We calculated the relationship between ISA percentage and NDVI value and plotted an ISA percentage as a function
of the NDVI map (Fig. 2). According to the statistical results, the negative correlation between ISA percentage and NDVI value does not fit well in a linear regression relationship. Under the linear regression assumption, there is an overestimation of ISA in the low-value range and underestimation of ISA in the high-value range. The logistic regression model (LRM) was selected for fitting (Walker and Duncan, 1967). Using this model, the ISA percentage and the maximum annual NDVI value fit better in high-value and low-value accumulation areas. In addition, the input parameters required by logistic regression—
ISA classification data and NDVI maximum data—can be obtained through existing methods and datasets (Weng, 2012; Gorelick et al., 2017). This relationship can be performed for data analysis. We conducted data analysis as follows: First, the annual NDVI maximum value and ISA classification data were retrieved from Landsat images. Second, the parameters of the logistic regression model were estimated. Third, the annual NDVI maximum value was used as input data to estimate the ISA fraction at the pixel level using the developed LRM as follows:

$$P(t) = \frac{1}{1+e^t} \tag{1}$$

$$t = a \times (1 - NDVI_{max}) + b, \tag{2}$$



where formula (1) refers to LRM, $a$ and $b$ refer to the parameters of LRM and $NDVI_{max}$ is the annual NDVI maximum value.

### 3.2 Retrieval of NDVI and ISA classification

### 3.2.1 Calculation of NDVI with Google Earth Engine

The annual NDVI maximum value is calculated from all NDVI images within one year using the maximum algorithm. During the process of combination, the impact of seasonal changes on classification is removed (Lu et al., 2008). Therefore, the vegetation percentage can be accurately extracted. The formula to calculate the annual NDVI maximum value is as follows:

$$NDVI_{max} = \max\{NDVI_1, NDVI_2, \cdots NDVI_i\}, \tag{3}$$

where $NDVI_{max}$ is the annual NDVI maximum value and $NDVI_i$ is the NDVI value of the $i$th image.

  We selected 30-m resolution Landsat images for calculating NDVI. All images were collected in Google Earth Engine (GEE) (Gorelick et al., 2017). GEE is a big-data processing platform for classification of multitemporal satellite imagery and geospatial data analysis. It integrates common remote sensing data such as Landsat, MODIS, Sentinel, DMSP-OLS, as well as thematic data such as population distribution and climate data. GEE performs by visual operation interface and analyzes

big data through JavaScript codes and functions for big data analyzes. In this study, all Landsat 5/7 images in 2000, 2005 and 2010 in urban areas and all Landsat 8 images in 2015 were selected to calculate the maximum value of NDVI in a given year.

  NDVI data were mainly distributed in the interval between 0 and 1 and stored in float format. Since this format is not conducive to data storage, processing and analysis, NDVI data were amplified by 1,000 and converted into integer format.

### 3.2.2 Extraction of ISA type for calibration

Urban ISA type was retrieved by linear spectral unmixing methods (Wu and Murray. 2003; Lu et al. 2014). The method was mainly divided into the following steps: First, based on the first three components using the minimum noise fraction (MNF) transformation, four endmembers—high-albedo surfaces (such as plazas, roofs), low-albedo surfaces (such as water, shadow), vegetation and bare soil—were selected. The spectral unmixing method was employed to unmix the Landsat multispectral bands into the four endmembers. A decision tree was built to classify the high-albedo surfaces, low-albedo

surfaces, water, vegetation and bare soil based on the fractions after unmixing and the calculation of indexes. The high-albedo and low-albedo surfaces were combined to generate the ISA type. We calculated NDVI and modified normalized difference water index (MNDWI) to remove the vegetation and water. The ISA classification of sample cities was extracted for calibration of parameters.



### 3.3 Retrieval of intra-urban structure/component

Due to regional differences in climate and geographical environments, huge discrepancies were found in the ISA and UGS components of cities in different regions in China. We chose sampling cities for the selection of sample points to calibrate the model. According to China's geographical zoning and urban development level, 28 large cities were selected for

calibrating the LRM's parameters (Fig. 3). LRM was calculated in each city with 1,000 random samples located in ISA and 1,000 random samples located in UGS or waterbody. These samples were merged to extract their corresponding annual NDVI maximum value. They were used as the input for LRM to calibrate its parameters (Fig. 4).

The mean values of parameters were calculated to obtain the LRM for each region (Table 2). Then, according to the LRMs in different regions, the annual NDVI maximum values generated by Landsat images were used as input to develop

the ISA fraction dataset for all sample cities (Fig. 5).

The waterbody data were derived from Landsat images of 2000, 2005, 2010 and 2015. The images between April and October were chosen. The modified normalized difference water index (MNDWI) was employed to delineate water distribution (Xu, 2006). Similar to the extraction of urban waterbody, the bare land data of 2000, 2005, 2010 and 2015 were derived from Landsat images as well. We used a simple index, enhanced built-up and bareness index (EBBI) for urban bare

land extraction (As-syakur et al., 2012). Because the spectral features of some green space are similar to bare land in non-growth season, it may influence the result of classification (Lu et al., 2008). In this study, the annual greenest-pixel top-of-atmosphere (TOA) reflectance composite products from GEE were used to EBBI calculation. The greenest-pixel means the pixel with the highest value of the NDVI, which remove seasonal turbulence on the classification of bare land. The distribution of water and bare land were extracted by threshold division, which was set by manual selection. Then we

retrieved the UGS fraction data. We assumed that in the urban area, the areas without water and bare land were mosaics of ISA and UGS. Among non-water and non-bare areas, the proportion of ISA was deducted, which generated the UGS fraction datasets.

### 3.4 Time-series data refinement

The spectral characteristics of unchanged urban areas as seen from multiple Landsat images varied, resulting in

inconsistency of ISA and UGS products. It was necessary, therefore, to refine the products before further analysis. The refinement was based on the following assumptions: (1) The generally increasing trend of impervious density inside Chinese cities is shown beside of urban greening area. (2) There is some unchanged area whose impervious density remained constant from year to year. (3) The UGS density may increase as a result of greening in local areas.

Because of its relatively high accuracy, the dataset of 2015 was chosen as a criterion for refinement of other period

datasets. The refinement process was conducted in this way: First, if the proportion of ISA in a certain pixel of a given year was lower than that in the previous year, it was modified to the proportion of ISA in that year as a precondition for those areas which did not implement greening projects.

The ISA fraction data was modified by the following formula:



$$X_{n-1} = \begin{cases} X_n & , if \ X_n \leq X_{n-1} \\ X_{n-1} & , if \ X_n > X_{n-1} \end{cases}, \tag{4}$$

where $X_{n-1}$ is the percentage of ISA in the former year and $X_n$ is the percentage of ISA in the later year.

Second, we generated random points inside an urban area for the year 2000. The sampling points where ISA or UGS fractions had changed were excluded in order to generate the unchanged points inside the city. For the second assumption, the proportion of ISA of some unchanged points was kept constant, because the proportion of ISA at these points should be consistent in all the periods.

Third, the proportion of ISA in unchanged points in each year was calculated. The proportion derived from the 2015 images was used to obtain the calibrating parameters for 2000, 2005 and 2010, which improved the initial ISA fraction data for those years.

## 4 Results and Discussion

### 4.1 Validation

#### 4.1.1 Validation of CLUDs

For each period—2000, 2005, 2010 and 2015—a unified quality check and data integration were performed to ensure the quality and consistency of the interpretation. In the process of remote sensing interpretation of land-use/coverage data, field investigations were mainly carried out in autumn in the northern part of the country and in spring in the southern part. High spatial resolution images from Google Earth have been commonly used for validation (Liu et al., 2014; Zhang et al., 2014; Kuang et al., 2016), and we adopted that method for this study. At least 2,200 random points for each interval were generated throughout China. Evaluating the results of comprehensive data quality in each interval, the overall accuracy of urban land or built-up area was more than 91.98% (Table 3).

### 4.1.2 Validation of ISA and UGS fractions

Google Earth images with higher spatial resolution than Landsat images were employed for the validation of ISA and UGS fractions. The 30 m × 30 m ISAs were first rectified with Google Earth images. A total of 1,111 validation samples with a window size of 3 × 3 pixels (90 m × 90 m grids) for each sample plot were acquired randomly from 44 cities in different regions in China and chosen for validation. We calculated the mean ISA and UGS density in each grid (Fig. 6). The actual value in the same area was obtained by visual interpretation from Google Earth images. Accuracy assessment of ISA and UGS was performed by root mean square error (RMSE) and R-square (Fig. 7). The RMSEs of ISA and UGS fractions were 0.10 and 0.14, respectively. The R-squares were 0.82 and 0.82, respectively.





## 4.2 Comparison of the CLUD-Urban product with other land-use products

We compared the vector boundaries of urban areas with the existing land-use products. There were obvious discrepancies among these products in data production, data source, resolution and definition of urban land-use types. The spatial resolutions of MODIS and ESA land-cover products range from 300 to 1000 m, and their classification systems are
based on IGBP and FAO frameworks, respectively (Belward, 1996; FAO, 1997). These products were selected for data comparison (Fig. 8).

The MODIS land-cover data uses the classification system defined by IGBP. In this system, the urban area is defined as built-up area, and is characterized by at least 30% ISA, including building materials and asphalt (Friedl et al., 2010). The ESA land-cover data considers urban areas to be artificial surfaces and associated areas (Bontemps et al., 2011). CLUD-
Urban distinguishes between urban and rural areas and can better characterize the urban boundaries (Liu et al., 2005).

We take Beijing as an example to demonstrate the different urban land-use products. The Landsat images acquired in 2000, 2005, 2010 and 2015 were used as reference (Fig. 8a), with a red-green-blue composition. The images of 2000, 2005 and 2010 were from Landsat 5 TM, and 2015 images were from Landsat 8 OLI. The resolution of MODIS land-cover products is relatively low (500 m) (Fig. 8c). Due to the coarse definition, the urban areas are relatively large. Moreover, there
was little change in urban coverage between different years, which did not allow characterization of urban expansion (Fig. 8c). Because both city and countryside are included in the MODIS definition of built-up area, this product cannot effectively distinguish between urban and rural lands. Although the ESA land-cover data had a better resolution of 300 m, urban and rural land still cannot be distinguished (Fig. 8d). In our dataset, the urban and rural areas are well distinguished because of a good definition of urban area (Fig. 8b).

## 4.3 Patterns of urban area change

To illustrate the general pattern of national urban land-use/cover change, we analyzed the process of urban expansion since 2000, together with ISA and UGS, which account large of urban area and carry ecological function in cities dynamics in China (Fig. 9, Fig. 10). The growth of ISA and UGS are obvious in main urban areas, like Beijing-Tianjin, Yangtze River Delta and Guangdong–Hong Kong–Macao Great Bay Area (Fig. 9, Fig. 10). Both ISA and UGS showed an increasing trend
associated with urban expansion. Higher proportions of ISA and UGS were located in eastern China, where economic conditions were better. Detailed fractions of ISA, UGS, waterbody and bare land in Shenyang are shown in Fig. 11, which illustrates urban internal composition. High proportional areas of ISA represent buildings, roads and plazas, whereas low proportional areas represent parks and greenbelts with ecological functions. This dataset can characterize differences among the selected cities. Some cities, like Beijing and Nanjing, with better-planned urban construction had smaller proportions of
ISA (64.60% and 68.19%, respectively) and higher proportions of UGS (33.81% and 30.33%, respectively) within their urban areas in 2015 (Fig. 12).

The total urban area increased from $3.26 \times 10^4 \, km^2$ in 2000 to $3.86 \times 10^4 \, km^2$ in 2005 to $6.28 \times 10^4 \, km^2$ in 2010 to $7.18 \times 10^4$ $km^2$ in 2015 (Fig. 13, Table 3). The five provinces with the largest urban land areas in 2015 were Shandong ($0.90 \times 10^4 \, km^2$),



Jiangsu ($0.81\times10^4$ km$^2$), Guangdong ($0.55\times10^4$ km$^2$), Hebei ($0.47\times10^4$ km$^2$) and Henan ($0.45\times10^4$ km$^2$). The results show that urban area in China was mainly distributed along the coast and in central regions. Although the western region accounts for the majority of China's total land area, its urban area is relatively small.

The regional differences were obvious among the provinces. The same five provinces also had the largest urban expansions between 2000 and 2015: Shandong ($0.58\times10^4$ km$^2$), Jiangsu ($0.52\times10^4$ km$^2$), Guangdong ($0.30\times10^4$ km$^2$), Hebei ($0.27\times10^4$ km$^2$) and Henan ($0.23\times10^4$ km$^2$). Four are located in the coastal region and one is in the central region, representing the two main urban expansion areas. With the implementation of China Western Development Plan in 2000 and Rise of Central China Plan in 2004, an acceleration of urban expansion also has been observed in western and other central provinces. For example, the urban expansion area of Anhui Province increased from 202.61 km$^2$ to 1,103.63 km$^2$ between 2000 and 2015, while adjacent Jiangsu Province increased its urban area slightly from 538.42 km$^2$ to 574.43 km$^2$ during the same period (Table 4).

## 4.4 Dynamics and patterns of ISA density

ISA, UGS, waterbody and bare land are main land-use types within cities. As a key component of urban structure, ISA growth was monitored during urban expansion. China's remotely sensed ISA was $2.22\times10^4$ km$^2$, $2.66\times10^4$ km$^2$, $4.30\times10^4$ km$^2$ and $5.10\times10^4$ km$^2$ in 2000, 2005, 2010 and 2015, respectively, showing an increasing trend (Fig. 14, Table 5), similar to the rate of expansion of urban area. From the perspective of the quality of dwellers' environment, the ISA density was 68.34%, 69.40%, 69.31% and 70.70% in 2000, 2005, 2010, and 2015, respectively (Fig. 15, Table 5). The results indicated that the ISA density in China showed a slight increasing trend. The percentage of ISA has stayed around 70%, showing a higher density ISA in urban area than other developed countries, like the USA (Kuang et al., 2014).

The spatiotemporal distribution of ISA in provinces across China was analyzed, and we found that it is mainly distributed in the coastal and the central regions (Fig. 14d). The distribution is relatively low in the western region. From 2000 to 2015, the ISA of each province showed an increasing trend. The three provinces with the fastest growing trend were Shandong ($0.45\times10^4$ km$^2$), Jiangsu ($0.37\times10^4$ km$^2$) and Guangdong ($0.23\times10^4$ km$^2$). Although the ISA in different provinces showed an increasing trend, the pattern of "high in east and low in west" remained unchanged in each year.

We further analyzed the urban ISA density in different provinces (Fig. 15). The ISA density in 2015 was employed for spatial characteristic delineation. Among all provinces, the three provinces with the lowest ISA density were Chongqing (56.54%), Taiwan (59.89%) and Heilongjiang (60.70%), which are located in southwestern, southeastern and northeastern China, respectively. This result showed that a more dispersed distribution of ISA may be due to reasons like climate, urban policy, etc. On the time scale, from 2000 to 2015, the ISA density showed an increasing trend in most provinces. However, the rate of increase was relatively small. The overall increase in urban ISA density from 2000 to 2015 was 2.36% (from 68.34% to 70.70%).



## 4.5 Changes in UGS

China's UGS monitored by remote sensing shows an increasing trend, similar to the trend of urban land area and urban ISA. The total UGS increased from $1.00 \times 10^4 \, \text{km}^2$ in 2000 to $1.15 \times 10^4 \, \text{km}^2$ in 2005 $1.90 \times 10^4 \, \text{km}^2$ in 2010 to $1.99 \times 10^4 \, \text{km}^2$ in 2015 (Fig. 16, Table 6). Looking at both ISA and UGS in urban areas, our results indicate a slight decrease in percentage of UGS, while ISA density has increased. The UGS fraction was 30.77%, 29.86%, 30.31% and 27.70%, in 2000, 2005, 2010 and 2015, respectively (Fig. 17, Table 6). The results show that the proportion of China's UGS decreased slightly; however, it has stayed around 30% at the national scale.

In order to analyze the regional differences in UGS distribution in China, the provincial administrative units were employed. As seen from the spatial distribution (Fig. 16), UGS is mainly distributed in coastal, northeastern, and southwestern China. From 2000 to 2015, the UGS in different provinces showed an increasing trend. The largest increase occurred in the coastal and northeastern regions. For example, UGS in the Jiangsu province increased by 1,676.03 km² from 2000 to 2015, reflecting better construction of UGS.

We further analyzed the percentages of UGS in different provinces (Fig. 17). The five provinces with the highest percentages of UGS were Chongqing (42.83%), Taiwan (39.22%), Heilongjiang (39.14%), Jilin (34.40%) and Beijing (33.11%), all located in eastern and northeastern China except Chongqing. This finding reflected better construction of UGS in the coastal and northeastern regions. From 2000 to 2015, the percentage of UGS in each province showed a decreasing trend. However, as a result of China's greening program, the percentage of UGS in some provinces has slightly increased. For example, after construction of UGS like Olympic Forest Park, the percentage of UGS in Beijing increased from 34.03% in 2000 to 34.21% in 2015.

## 5 Data availability

All data presented in this paper are available in https://doi.org/10.5281/zenodo.2644932 (Kuang et al., 2019). This datasets cover 2000, 2005, 2010 and 2015 with a spatial resolution of 30m. Detailed metadata are associated in the introduction, including contact information.

## 6 Conclusion

In this study, CLUD-Urban, a 30-m resolution national urban land-cover change data series for China, was generated using a satellite big-data platform. CLUD-Urban was able to distinguish detailed urban land classification and urban internal structures for 2000, 2005, 2010 and 2015. The novelty of this dataset, compared to previous products, is that it regards cities as heterogeneous, consisting of ISA, UGS, waterbody and bare land. Due to the integration of visual interpretation with prior knowledge, the accuracy of the CLUD-Urban dataset is more than 91.98% for remotely sensed urban areas. The logistic regression relationship between the annual maximum NDVI value and urban ISA was built at the city scale. We developed a





time series of urban land-cover change, including ISA, UGS, which can be used to accurately characterize the urban internal structure. The RMSEs of ISA and UGS fractions are 0.10 and 0.14, respectively. Results from the analysis of urban areas, including ISA and UGS and their proportions for each province, show large regional differences in China.

## Author contribution

5     KW, ZS and LX designed the research; ZS and LX implemented the research; KW and ZS wrote the paper.

## Competing interests

The authors declare that they have no conflict of interest.

## Acknowledgments

This study was supported by Strategic Priority Research Program A of the Chinese Academy of Sciences
10   (XDA23100201) and National Natural Science Foundation of China (NSFC) (41871343). We thank Dr. Sun Fengyun and Dr. Rafiq Hamdi for their help in manuscript revising.

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



**Table 1**: **Characteristics of the multitemporal data series**

| Path | Row | Year | Sensor | Spatial resolution (m) |
|---|---|---|---|---|
| 118–149 | 23–43 | 1999–2001 | TM, ETM+ | 30 |
| | | 2004–2006 | TM | 30 |
| | | 2009–2011 | TM, HJ-1, CBERS -1 | 30 |
| | | 2014–2016 | OLI | 30 |

Note: TM, Landsat Thematic Mapper; ETM+, Landsat Enhanced Thematic Mapper Plus; HJ, Huan Jing; CBERS, China Brazil Earth Resources Satellite; OLI, Landsat 8 Operational Land Imager.

**Table 2**: **Parameters of the logistic regression model of sample cities in China.**

| City | a | b | Score |
|---|---|---|---|
| Changsha | 0.013 | -6.353 | 0.866 |
| Hefei | 0.011 | -6.148 | 0.855 |
| Taiyuan | 0.014 | -7.974 | 0.873 |
| Wuhan | 0.008 | -4.425 | 0.802 |
| Central region | 0.012 | -6.225 | 0.849 |
| Tianjin | 0.007 | -3.946 | 0.780 |
| Haikou | 0.012 | -6.357 | 0.897 |
| Jinan | 0.015 | -9.149 | 0.879 |
| Hangzhou | 0.010 | -5.032 | 0.887 |
| Nanjing | 0.006 | -3.503 | 0.768 |
| Shenzhen | 0.011 | -5.891 | 0.873 |
| Qingdao | 0.013 | -8.399 | 0.845 |
| Xiamen | 0.009 | -5.224 | 0.833 |
| Ningbo | 0.010 | -4.988 | 0.862 |
| Foshan | 0.007 | -3.957 | 0.757 |
| Dongguan | 0.006 | -3.904 | 0.792 |
| Beijing | 0.010 | -5.311 | 0.876 |
| Chengdu | 0.012 | -6.669 | 0.834 |
| Shanghai | 0.008 | -4.362 | 0.784 |
| Coastal region | 0.010 | -5.478 | 0.833 |
| Changchun | 0.013 | -6.054 | 0.893 |
| Harbin | 0.012 | -5.569 | 0.939 |
| Shenyang | 0.011 | -5.370 | 0.848 |
| Dalian | 0.012 | -7.240 | 0.881 |
| Northeastern region | 0.012 | -6.058 | 0.890 |
| Guiyang | 0.014 | -7.902 | 0.884 |
| Kunming | 0.013 | -7.074 | 0.878 |
| Nanning | 0.015 | -0.722 | 0.913 |
| Xian | 0.014 | -7.571 | 0.911 |



| City | a | b | Score |
|---|---|---|---|
| Lanzhou | 0.017 | -12.357 | 0.827 |
| Urumqi | 0.003 | -2.058 | 0.686 |
| Western region | 0.013 | -6.281 | 0.850 |

Note: a, b are parameters of LRM

**Table 3: Accuracy assessments from extracted urban areas.**

| Year | Land types | Accuracy analysis for land types | | | |
|---|---|---|---|---|---|
| | | PA (%) | UA (%) | Overall accuracy | |
| 2000 | Built-up area | − | − | 98.92% | (Zhang et al., 2014) |
| 2005 | Built-up area | − | − | 97.01% | (Zhang et al., 2014) |
| 2010 | Urban land | 94.3 | 93.67 | 93.99% | (Kuang et al., 2016) |
| 2015 | Urban land | 91.3 | 92.65 | 91.98% | (Zhang et al., 2019) |

Note: PA, producer's accuracy; UA, user's accuracy.

**Table 4: Urban area of each province in China (km².).**

| Province | Urban Area (km²) | | | | Area of expansion (km²) | | | |
|---|---|---|---|---|---|---|---|---|
| | 2000 | 2005 | 2010 | 2015 | 2000–2005 | 2005–2010 | 2010–2015 | 2000–2015 |
| Anhui | 1015.70 | 1218.31 | 1920.28 | 3023.91 | 202.61 | 701.97 | **1103.63** | 2008.21 |
| Beijing | 1034.95 | 1300.46 | 1493.38 | 2021.48 | 265.51 | 192.92 | 528.10 | 986.53 |
| Chongqing | 328.21 | 394.72 | 695.57 | 763.17 | 66.51 | 300.85 | 67.60 | 434.96 |
| Fujian | 657.39 | 1247.52 | 1273.17 | 1487.85 | 590.12 | 25.65 | 214.69 | 830.46 |
| Gansu | 406.15 | 486.67 | 529.78 | 793.94 | 80.53 | 43.11 | 264.16 | 387.80 |
| Guangdong | 2468.96 | **4231.81** | 4830.30 | 4966.04 | **1762.85** | 598.49 | 135.74 | 2497.08 |
| Guangxi | 826.08 | 927.64 | 1079.28 | 1179.99 | 101.56 | 151.64 | 100.70 | 353.90 |
| Guizhou | 251.53 | 262.97 | 416.85 | 592.42 | 11.44 | 153.88 | 175.57 | 340.89 |
| Hainan | 176.12 | 226.38 | 265.31 | 326.76 | 50.26 | 38.93 | 61.46 | 150.65 |
| Hebei | 1929.96 | 2375.46 | 2707.50 | 2906.56 | 445.50 | 332.05 | 199.06 | 976.61 |
| Heilongjiang | 1366.89 | 1386.11 | 1493.71 | 1664.27 | 19.22 | 107.60 | 170.57 | 297.38 |
| Henan | 2161.61 | 2768.73 | 3108.40 | 3495.12 | 607.12 | 339.67 | 386.72 | 1333.51 |
| Hong Kong | 158.29 | 162.27 | 163.62 | 164.48 | 3.98 | 1.35 | 0.86 | 6.19 |
| Hubei | 1158.35 | 1314.33 | 1458.59 | 2556.43 | 155.98 | 144.26 | 1097.84 | 1398.08 |
| Hunan | 956.38 | 1135.31 | 1297.28 | 1874.85 | 178.93 | 161.96 | 577.57 | 918.47 |
| Inner Mongolia | 1162.29 | 1313.52 | 1406.42 | 1921.78 | 151.23 | 92.90 | 515.36 | 759.49 |
| Jiangsu | 2833.86 | 3372.28 | 4444.52 | 5018.95 | 538.42 | 1072.24 | 574.43 | 2185.09 |
| Jiangxi | 584.63 | 849.22 | 978.17 | 1071.28 | 264.59 | 128.95 | 93.11 | 486.65 |



| Province | Urban Area (km²) | | | | Area of expansion (km²) | | | |
|---|---|---|---|---|---|---|---|---|
| | 2000 | 2005 | 2010 | 2015 | 2000–2005 | 2005–2010 | 2010–2015 | 2000–2015 |
| Jilin | 1019.47 | 1096.78 | 1223.93 | 1450.30 | 77.32 | 127.15 | 226.37 | 430.83 |
| Liaoning | 1576.10 | 1648.93 | 1843.61 | 1937.24 | 72.83 | 194.68 | 93.63 | 361.14 |
| Macao | 6.82 | 6.93 | 6.96 | 7.11 | 0.11 | 0.03 | 0.15 | 0.29 |
| Ningxia | 135.71 | 205.07 | 274.16 | 347.57 | 69.36 | 69.09 | 73.41 | 211.85 |
| Qinghai | 129.43 | 147.22 | 154.91 | 177.68 | 17.79 | 7.68 | 22.77 | 48.24 |
| Shaanxi | 551.48 | 663.71 | 748.22 | 924.38 | 112.23 | 84.52 | 176.15 | 372.90 |
| Shandong | **3140.36** | 4065.78 | **5178.11** | **5657.86** | 925.42 | **1112.33** | 479.75 | **2517.50** |
| Shanghai | 777.94 | 914.15 | 1031.53 | 1078.69 | 136.22 | 117.38 | 47.16 | 300.76 |
| Shanxi | 985.76 | 1121.33 | 1334.22 | 1370.87 | 135.57 | 212.89 | 36.65 | 385.12 |
| Sichuan | 904.44 | 1176.37 | 1568.16 | 1686.28 | 271.93 | 391.79 | 118.12 | 781.84 |
| Taiwan | 1158.95 | 1253.50 | 1270.90 | 1394.69 | 94.55 | 17.40 | 123.79 | 235.74 |
| Tianjin | 577.30 | 839.94 | 1179.49 | 1306.01 | 262.64 | 339.55 | 126.52 | 728.71 |
| Tibet | 52.83 | 73.14 | 77.73 | 108.93 | 20.31 | 4.58 | 31.20 | 56.10 |
| Xinjiang | 1029.57 | 1182.44 | 1250.51 | 1910.37 | 152.87 | 68.07 | 659.87 | 880.80 |
| Yunnan | 522.71 | 603.61 | 1083.52 | 1147.71 | 80.91 | 479.91 | 64.20 | 625.01 |
| Zhejiang | 1149.66 | 2403.59 | 2484.72 | 2957.29 | 1253.93 | 81.14 | 472.57 | 1807.63 |
| Overall | 35195.86 | 44381.20 | 52282.80 | 61307.27 | 9180.34 | 7896.59 | 9019.47 | 26096.40 |

Note: The largest urban areas and areas of expansion are shown in **bold**.

**Table 5: Urban impervious surface area (ISA) of China at provincial scale.**

| Province | Feature of ISA in 2000 | | Area of ISA expansion (km²) | | | Feature of ISA in 2015 | |
|---|---|---|---|---|---|---|---|
| | Percentage | Area (km²) | 2000-2005 | 2005-2010 | 2010-2015 | Percentage | Area (km²) |
| Anhui | 66.74% | 677.88 | 149.24 | 425.29 | **766.11** | 66.75% | 2018.53 |
| Beijing | 64.78% | 670.49 | 168.19 | 142.53 | 324.76 | 64.60% | 1305.96 |
| Chongqing | 56.61% | 185.81 | 50.99 | 146.06 | 48.65 | 56.54% | 431.51 |
| Fujian | 68.89% | 452.91 | 353.40 | 56.02 | 167.34 | 69.20% | 1029.67 |
| Gansu | 82.95% | 336.89 | 63.75 | 39.15 | 224.09 | 83.62% | 663.89 |
| Guangdong | 71.36% | 1761.90 | **1212.82** | 608.08 | 109.91 | 74.36% | 3692.71 |
| Guangxi | 60.98% | 503.72 | 83.18 | 149.66 | 84.05 | 69.54% | 820.62 |
| Guizhou | 68.71% | 172.82 | 17.81 | 108.78 | 153.91 | 76.52% | 453.32 |
| Hainan | 55.06% | 96.97 | 32.58 | 43.66 | 46.23 | 67.16% | 219.44 |
| Hebei | 70.52% | 1361.03 | 292.94 | 181.31 | 269.60 | 72.42% | 2104.87 |
| Heilongjiang | 60.93% | 832.84 | 27.25 | 8.53 | 141.62 | 60.70% | 1010.23 |
| Henan | 68.31% | 1476.57 | 439.19 | 241.91 | 356.02 | 71.92% | 2513.69 |
| Hong Kong | 64.94% | 102.79 | 3.21 | 0.94 | 0.25 | 65.17% | 107.19 |
| Hubei | 63.33% | 733.61 | 105.73 | 124.36 | 712.94 | 65.59% | 1676.64 |



| Province | Feature of ISA in 2000 | | Area of ISA expansion (km²) | | | Feature of ISA in 2015 | |
|---|---|---|---|---|---|---|---|
| | Percentage | Area (km²) | 2000-2005 | 2005-2010 | 2010-2015 | Percentage | Area (km²) |
| Hunan | 68.27% | 652.89 | 140.18 | 152.06 | 405.81 | 72.06% | 1350.93 |
| Inner Mongolia | 77.50% | 900.78 | 129.07 | 109.41 | 401.66 | 80.18% | 1540.92 |
| Jiangsu | 60.44% | 1712.85 | 434.16 | 528.81 | 713.05 | 67.52% | 3388.88 |
| Jiangxi | 65.96% | 385.65 | 184.57 | 121.47 | 67.10 | 70.83% | 758.79 |
| Jilin | 63.64% | 648.80 | 61.89 | 60.66 | 171.48 | 65.01% | 942.83 |
| Liaoning | 70.01% | 1103.48 | 74.83 | 87.32 | 105.62 | 70.78% | 1371.26 |
| Macao | 76.21% | 5.20 | 0.28 | 0.18 | 0.24 | 83.05% | 5.90 |
| Ningxia | 84.89% | 115.21 | 56.36 | 50.97 | 63.34 | 82.25% | 285.88 |
| Qinghai | 73.09% | 94.61 | 13.79 | 9.19 | 28.96 | 82.48% | 146.55 |
| Shaanxi | 71.29% | 393.16 | 90.72 | 71.76 | 141.76 | 75.45% | 697.40 |
| Shandong | 69.63% | **2186.57** | 729.50 | **725.61** | 549.14 | 74.07% | **4190.82** |
| Shanghai | 75.69% | 588.81 | 124.93 | 99.12 | 6.29 | 75.94% | 819.14 |
| Shanxi | 78.84% | 777.17 | 115.07 | 74.52 | 82.57 | 76.54% | 1049.33 |
| Sichuan | 67.17% | 607.52 | 181.53 | 290.37 | 89.78 | 69.34% | 1169.21 |
| Taiwan | 56.21% | 651.47 | 80.67 | 23.30 | 79.89 | 59.89% | 835.33 |
| Tianjin | 74.90% | 432.43 | 180.98 | 298.06 | 70.35 | 75.18% | 981.81 |
| Tibet | 84.06% | 44.41 | 12.54 | 3.54 | 30.36 | 83.40% | 90.85 |
| Xinjiang | 79.08% | 814.14 | 55.97 | 94.42 | 682.82 | 80.61% | 1647.35 |
| Yunnan | 59.64% | 311.72 | 61.80 | 341.17 | 81.10 | 69.34% | 795.79 |
| Zhejiang | 70.76% | 813.44 | 831.19 | 122.29 | 389.26 | 72.91% | 2156.18 |
| Overall | 68.34% | 22606.54 | 6560.32 | 5540.51 | 7566.05 | 70.70% | 42273.41 |

Note: The highest provincial expansions are shown in **bold**.

**Table 6: Urban green space (UGS) in China at provincial scale.**

| Province | Feature of UGS in 2000 | | Area of UGS expansion (km²) | | | Feature of UGS in 2015 | |
|---|---|---|---|---|---|---|---|
| | Percentage | Area (km²) | 2000-2005 | 2005-2010 | 2010-2015 | Percentage | Area (km²) |
| Anhui | 32.28% | 327.82 | 51.38 | 269.77 | 326.65 | 32.26% | 975.62 |
| Beijing | 34.03% | 352.18 | 94.16 | 48.10 | 197.07 | 34.21% | 691.52 |
| Chongqing | 42.75% | 140.33 | 15.10 | 152.88 | 18.52 | 42.83% | 326.83 |
| Fujian | 28.19% | 185.30 | 219.50 | -31.12 | 41.08 | 27.88% | 414.77 |
| Gansu | 16.26% | 66.03 | 16.13 | 3.62 | 37.97 | 15.59% | 123.74 |
| Guangdong | 26.04% | 642.87 | **504.20** | -25.16 | 22.30 | 23.04% | 1144.21 |
| Guangxi | 35.88% | 296.44 | 15.19 | -2.78 | 13.49 | 27.32% | 322.34 |
| Guizhou | 29.45% | 74.09 | -6.58 | 42.27 | 18.44 | 21.64% | 128.22 |
| Hainan | 44.51% | 78.40 | 17.46 | -4.90 | 14.96 | 32.41% | 105.92 |
| Hebei | 28.65% | 553.01 | 148.88 | 148.00 | -72.18 | 26.76% | 777.72 |





| Province | Feature of UGS in 2000 | | Area of UGS expansion (km²) | | | Feature of UGS in 2015 | |
|---|---|---|---|---|---|---|---|
| | Percentage | Area (km²) | 2000-2005 | 2005-2010 | 2010-2015 | Percentage | Area (km²) |
| Heilongjiang | 38.91% | 531.91 | -8.07 | 127.19 | 0.39 | 39.14% | 651.42 |
| Henan | 30.84% | 666.62 | 162.75 | 94.87 | 27.40 | 27.23% | 951.64 |
| Hong Kong | 31.41% | 49.72 | 0.63 | 0.36 | 0.58 | 31.19% | 51.30 |
| Hubei | 34.99% | 405.30 | 47.64 | 17.48 | **366.48** | 32.74% | 836.89 |
| Hunan | 28.96% | 276.99 | 33.80 | 5.42 | 155.76 | 25.17% | 471.97 |
| Inner Mongolia | 22.36% | 259.87 | 21.95 | -16.65 | 112.97 | 19.68% | 378.13 |
| Jiangsu | 38.21% | **1082.83** | 97.00 | **528.98** | -146.36 | 31.13% | **1562.45** |
| Jiangxi | 32.12% | 187.81 | 74.96 | 5.01 | 24.23 | 27.26% | 292.01 |
| Jilin | 35.76% | 364.60 | 14.97 | 65.73 | 53.54 | 34.40% | 498.84 |
| Liaoning | 29.72% | 468.38 | -2.19 | 106.83 | -12.24 | 28.95% | 560.78 |
| Macao | 5.19% | 0.35 | -0.20 | -0.15 | -0.01 | 0.08% | 0.01 |
| Ningxia | 14.84% | 20.14 | 12.80 | 17.93 | 9.87 | 17.48% | 60.74 |
| Qinghai | 26.74% | 34.61 | 3.97 | -1.52 | -6.23 | 17.35% | 30.83 |
| Shaanxi | 28.32% | 156.19 | 21.07 | 12.43 | 33.71 | 24.17% | 223.40 |
| Shandong | 29.81% | 936.22 | 190.75 | 380.51 | -72.08 | 25.37% | 1435.41 |
| Shanghai | 22.87% | 177.93 | 9.33 | 16.56 | 40.20 | 22.62% | 244.02 |
| Shanxi | 20.88% | 205.78 | 20.11 | 137.77 | -46.02 | 23.17% | 317.64 |
| Sichuan | 31.02% | 280.57 | 85.48 | 94.34 | 26.21 | 28.86% | 486.61 |
| Taiwan | 42.90% | 497.18 | 13.04 | -6.05 | 42.80 | 39.22% | 546.97 |
| Tianjin | 23.42% | 135.19 | 77.25 | 35.80 | 54.05 | 23.15% | 302.29 |
| Tibet | 12.62% | 6.67 | 7.50 | 0.92 | -0.02 | 13.83% | 15.07 |
| Xinjiang | 20.70% | 213.09 | 95.02 | -24.37 | -21.94 | 13.70% | 261.80 |
| Yunnan | 39.68% | 207.42 | 18.56 | 135.46 | -17.34 | 29.98% | 344.10 |
| Zhejiang | 27.10% | 311.53 | 395.81 | -42.89 | 73.16 | 24.94% | 737.62 |
| Overall | 28.96% | 10193.34 | 2469.39 | 2292.66 | 1317.42 | 26.54% | 16272.81 |

Note: The highest provincial expansions are shown in **bold**.



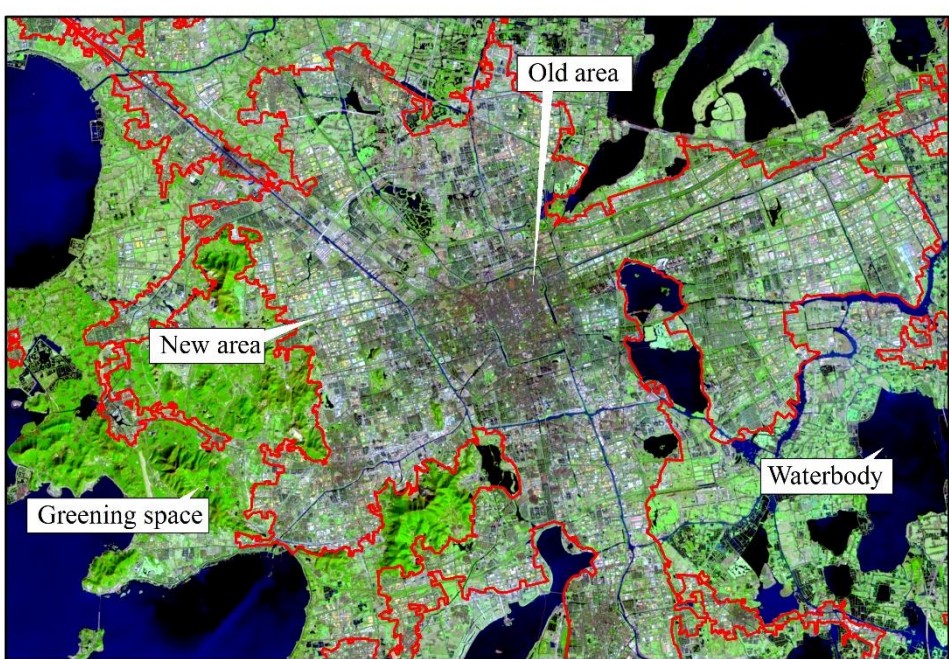

**Figure 1: Landsat image and digital urban boundary (in red) of Suzhou, China (December 29, 2014, Landsat 8 OLI, composite of Short-Wave-Infrared, Near-Infrared and Red bands). (The Landsat image is provided by Geospatial Data Cloud site, Computer Network Information Center, Chinese Academy of Sciences (http://www.gscloud.cn))**

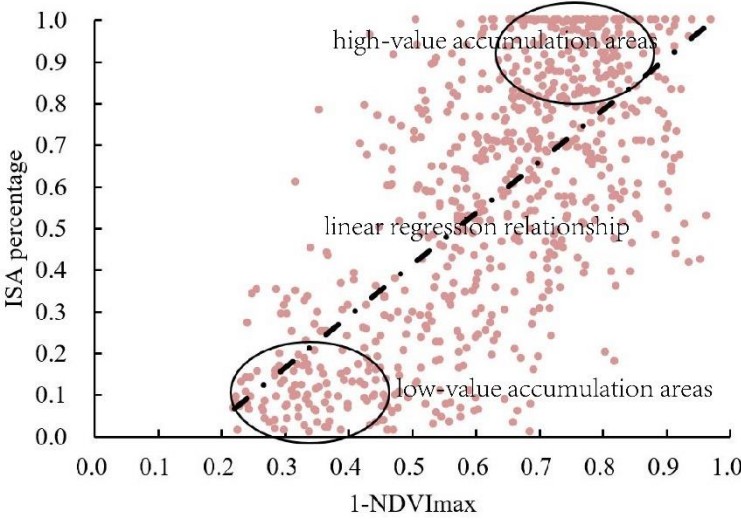

**Figure 2: The relationship between 1 sub NDVImax (1-NDVImax) and impervious surface area (ISA) percentage of pixels.**



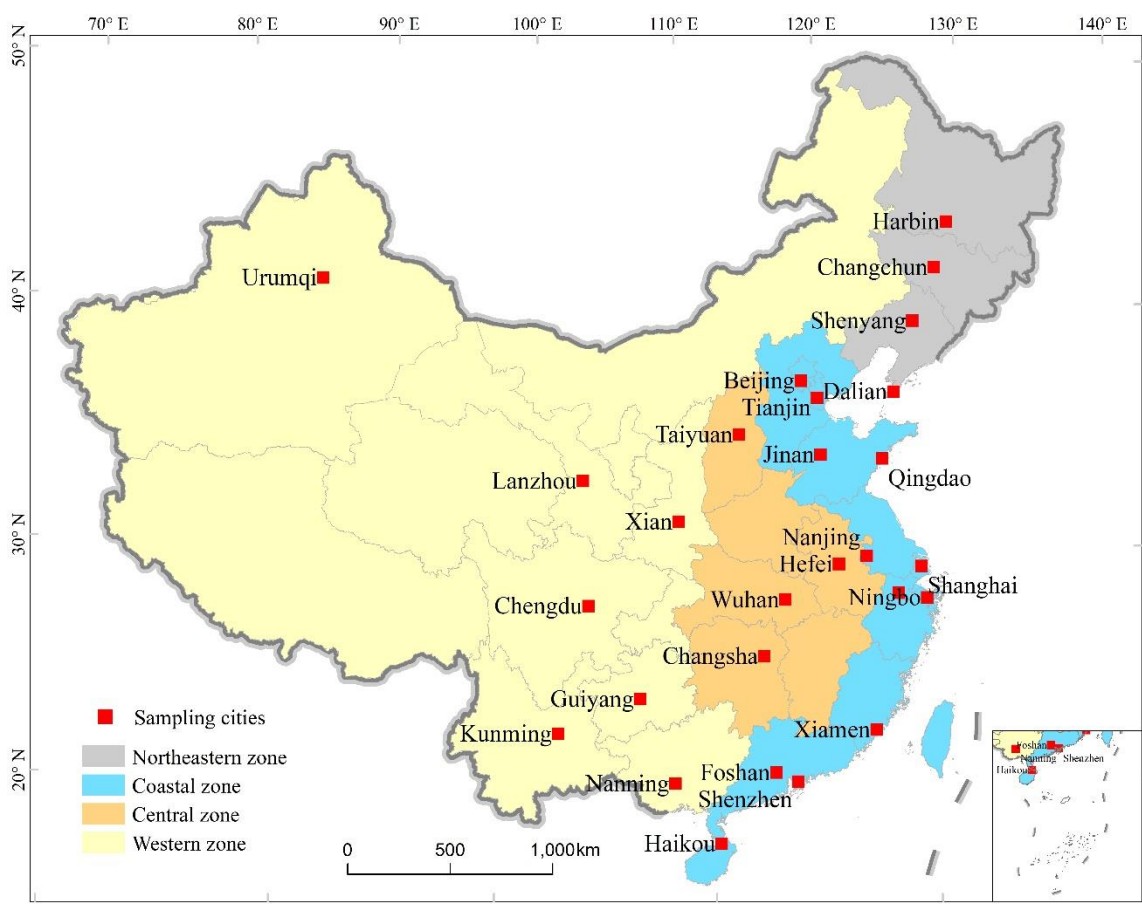

**Figure 3**: **Spatial distribution of 28 sample cities located in different regions of China.**

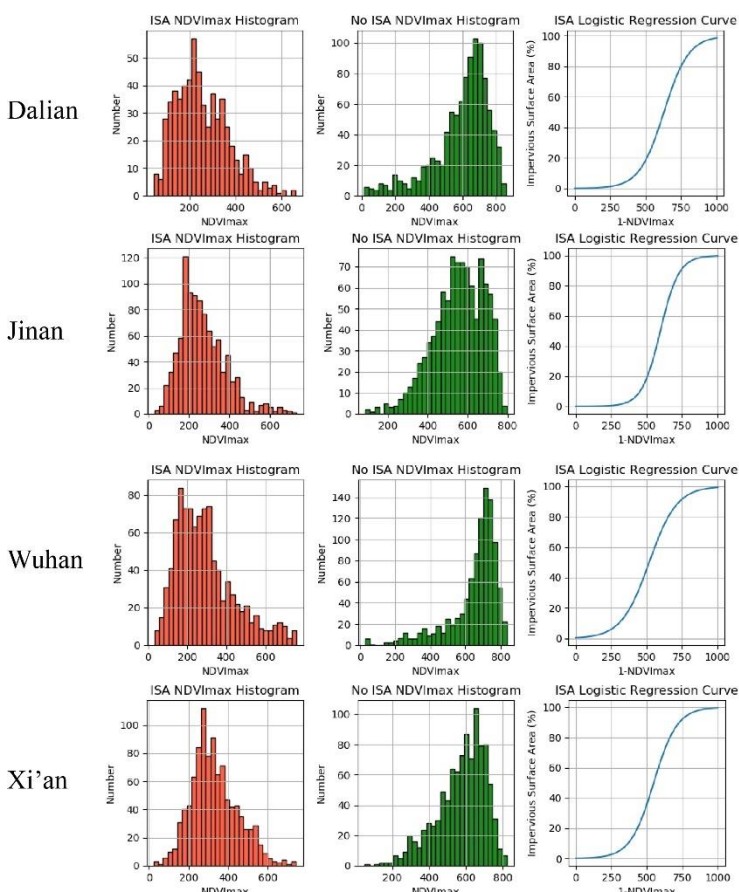

**Figure 4**: **Relationships between ISAs and annual NDVI maximum values in four sample cities in China.**

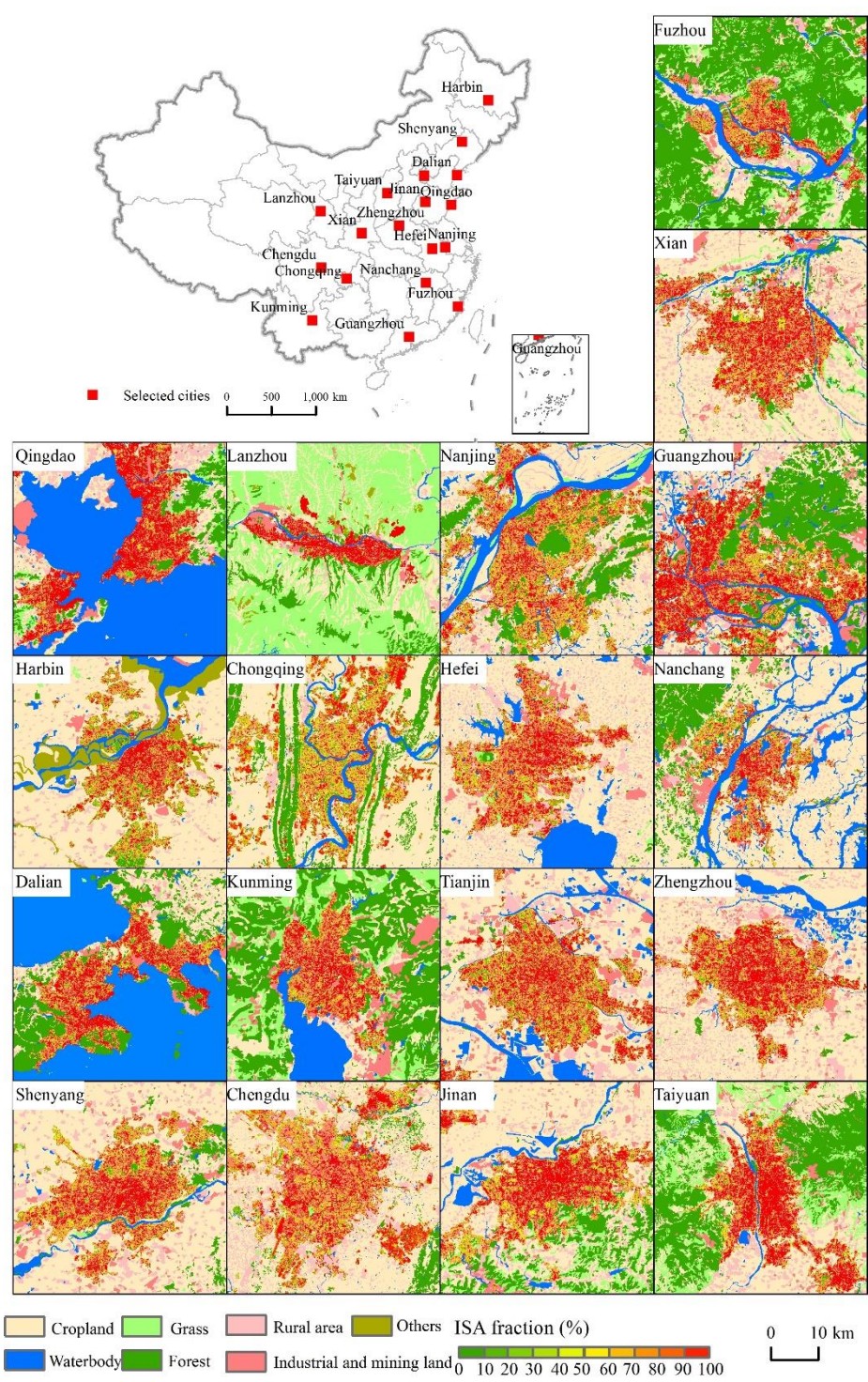

**Figure 5**: **Urban land-use/cover mapping of selected cities in China.**



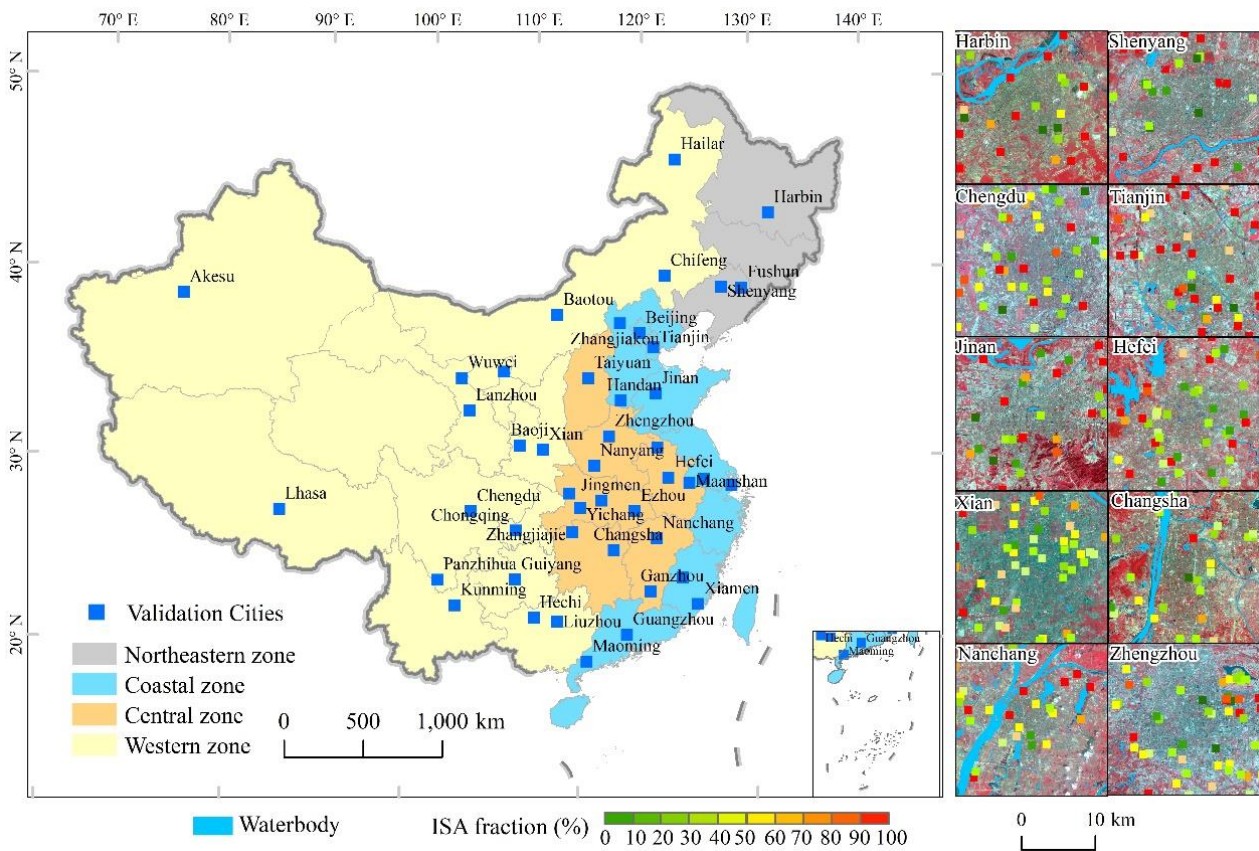

**Figure 6**: **Distribution of 44 validation cities and validation points in China. (The image is provided by Geospatial Data Cloud site, Computer Network Information Center, Chinese Academy of Sciences (http://www.gscloud.cn))**

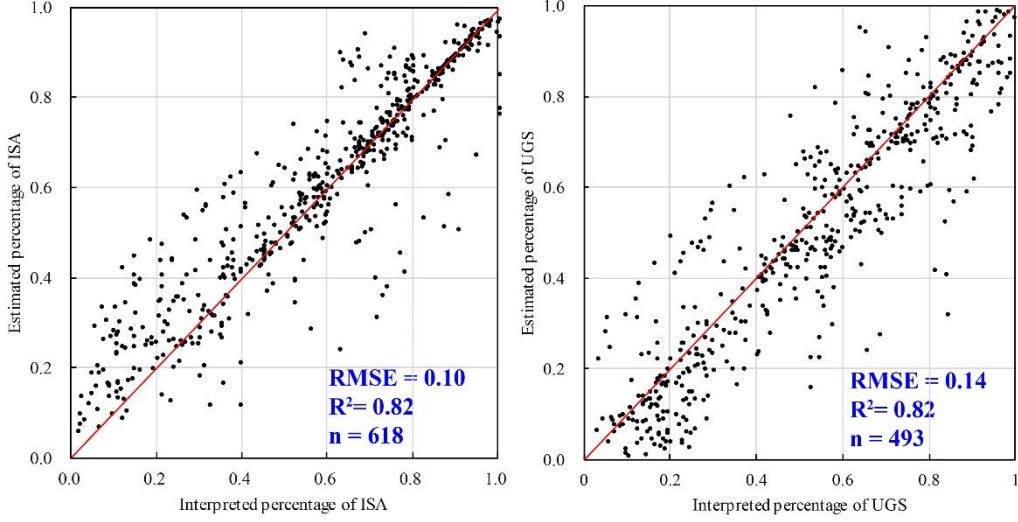

**Figure 7**: **Validation of remotely sensed impervious surface area (ISA) and urban green space (UGS) fractions.**

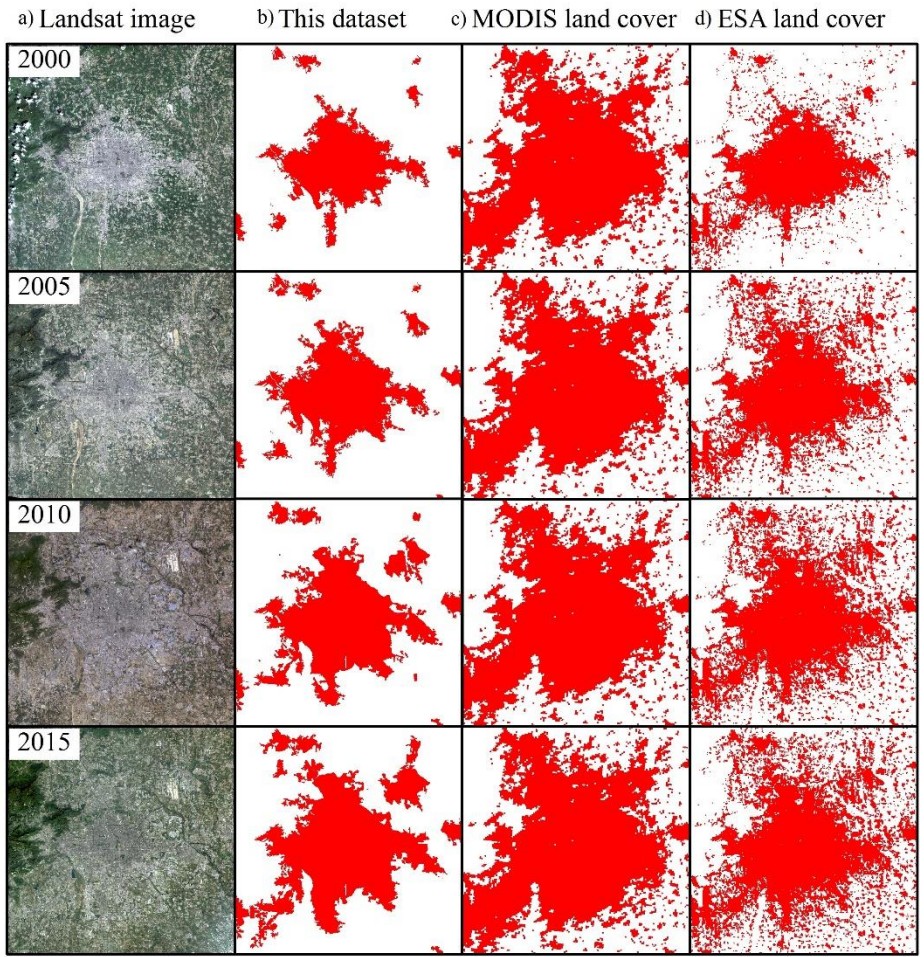

**Figure 8**: **Comparison of urban land classifications of Beijing, China, in three urban land-use products. (The Landsat image is provided by Geospatial Data Cloud site, Computer Network Information Center, Chinese Academy of Sciences**

5    **(http://www.gscloud.cn))**



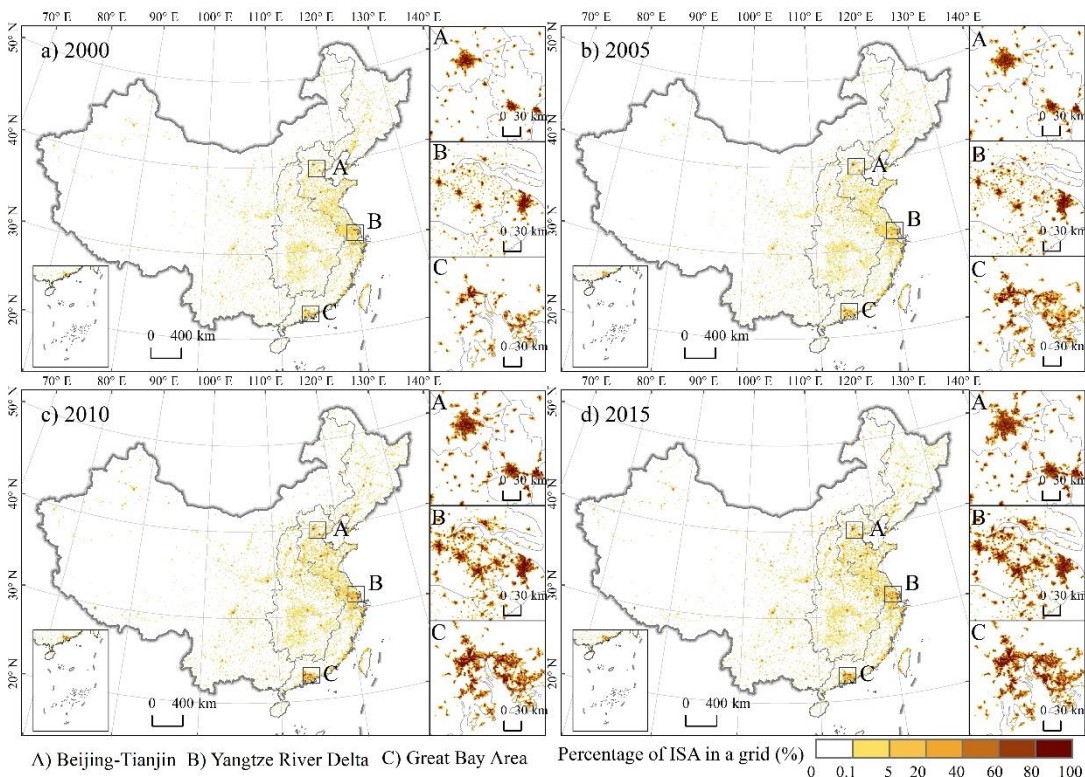

**Figure 9**: **Spatiotemporal change in impervious surface area (ISA) in China, 2000–2015 at five-year intervals.**

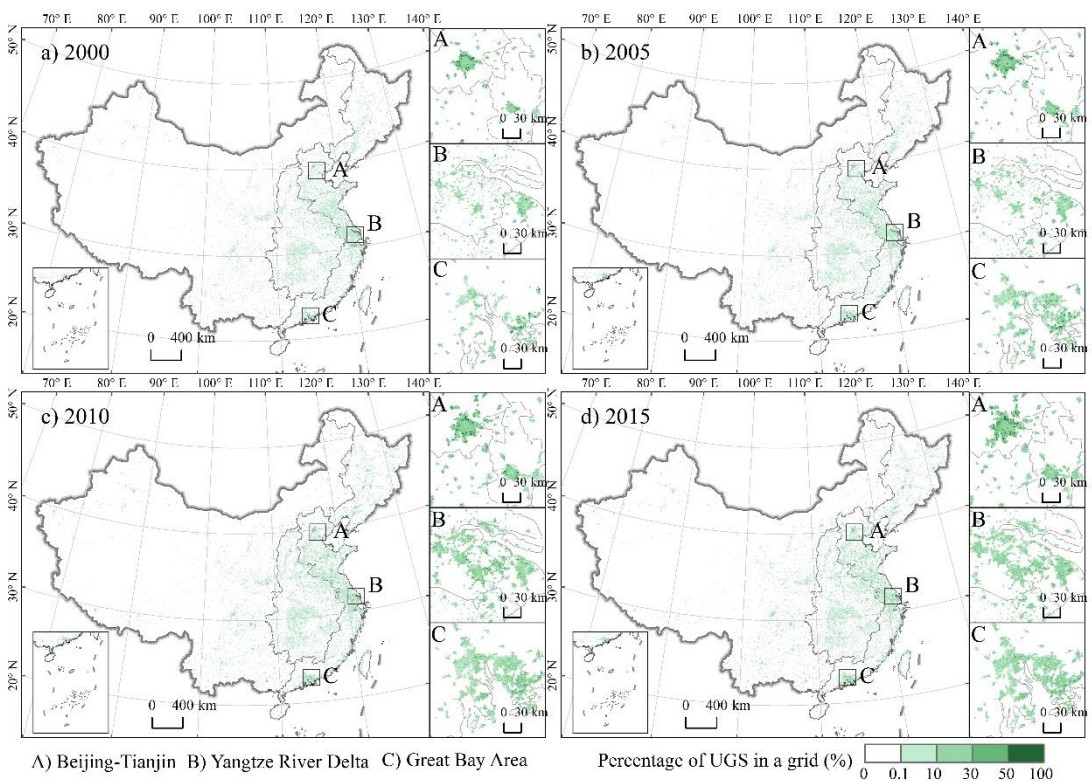

**Figure 10**: **Spatiotemporal change in urban green space (UGS) in China, 2000–2015 at five-year intervals.**

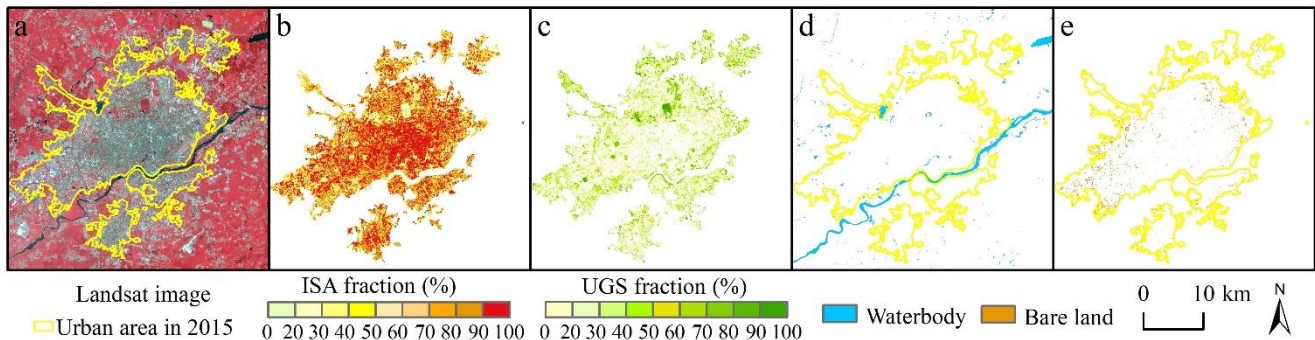

**Figure 11**: **Urban components of impervious surface area (ISA), urban green space (UGS), waterbody and bare land in Shenyang, China, 2015 (Landsat 8 OLI image, with false color composition). (The Landsat image is provided by Geospatial Data Cloud site, Computer Network Information Center, Chinese Academy of Sciences (http://www.gscloud.cn))**



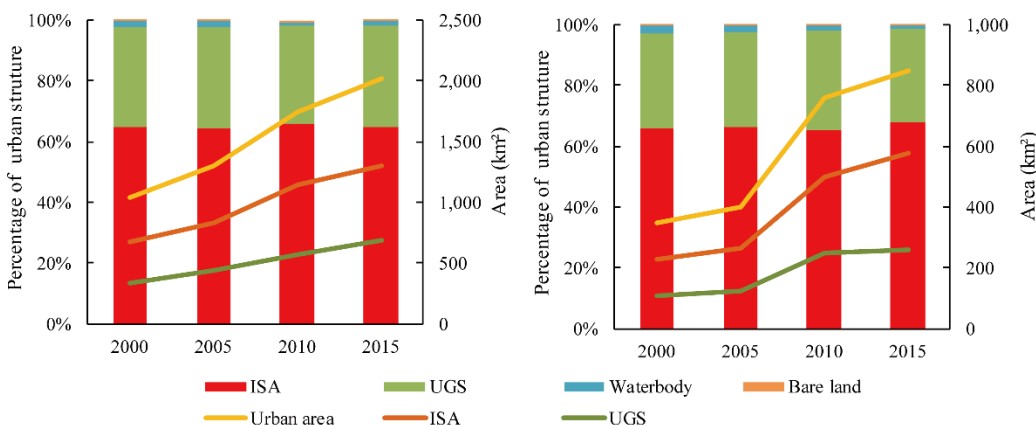

**Figure 12**: **Urban land change in Beijing and Nanjing, China, 2000–2015 at five-year intervals.**

a) 2000

b) 2005

c) 2010

d) 2015

Urban area (km²)

0 - 500

500 - 1000

1000 - 1500

1500 - 2000

2000 - 4000

> 4000

**Figure 13**: **The amounts of urban area in China's provinces in 2000, 2005, 2010 and 2015.**



a) 2000

b) 2005

c) 2010

d) 2015

ISA (km²)

0 - 500
500 - 1000
1000 - 1500
1500 - 2000
2000 - 4000
> 4000

**Figure 14: Impervious surface area (ISA) in each province in China for 2000, 2005, 2010 and 2015.**

a) 2000

b) 2005

c) 2010

d) 2015

Percentage of ISA (%)

60  65  70  75  80

**Figure 15: The percentage of impervious surface area (ISA) in each province in China for 2000, 2005, 2010 and 2015.**



a) 2000

b) 2005

c) 2010

d) 2015

Area of UGS (km$^2$)

- 0 - 100
- 100 - 200
- 200 - 500
- 500 - 1000
- 1000 - 1500
- >1500

0    500 km

**Figure 16: Urban green space (UGS) in each province in China for 2000, 2005, 2010 and 2015.**

**Figure 17: Percentage of urban green space (UGS) in each province in China for 2000, 2005, 2010 and 2015.**

