# Peer review of "A 30-meter resolution national urban land-cover dataset of China, 2000–2015"

_Earth System Science Data, 2019_

## Short Comment (SC1) · 12 Sep 2019

A 30-meter resolution national urban land-cover data-set of China, 2000–2015 will give opportunity to conduct interesting research related to the topic of urban climate and the contribution of the land use land cover change including historical urbanization in the climate evolution at regional scale.

---

## Referee Comment (RC1) · Anonymous Referee #1 · 13 Sep 2019

Summary and general comments

Kuang et al. refined an existing land use and land cover data set (China's Land Use/cover Dataset) specifically for generating fractions of impervious surface area (ISA) and vegetation (within cities) at national level. While I find the manuscript and the dataset of general interest, I still have some concerns that I think the authors need further consideration. I will comment on this manuscript following the guidelines from the publisher's website.

1. Are the data and methods presented new? Is there any potential of the data being useful in the future? Are methods and materials described in sufficient detail? Are any references/citations to other data sets or articles missing or inappropriate?

The data set related to ISA fraction and urban vegetation fraction at national level presented in the manuscript is new but not the method used to estimate them. The use of NDVI and other auxiliary data including reflectance to estimate ISA fraction has been done previously (e.g., Sexton et al., 2013, Remote Sensing of Environment). Obviously, these types of citations are neglected in the manuscript. Another factor that may lead to the judgement of the data set presented in this manuscript not as useful as it claimed by the authors is the mapping interval (5-year). A quick search of the current literatures would tell you that the scientific community now advocates for urban land cover dataset at a higher temporal frequency (e.g., annual mapping) particularly for urban environmental and climate studies. However, the authors did not even identify/mention possible use of their datasets of a five-year interval. For example, how does your dataset contribute to "world urban database" that may eventually help studies in urban climate using earth system models (e.g., weather research and forecasting, WRF)? I am not advocating for a case study or specifically linking your dataset to "world urban database", but a potential linkage between this presented dataset and environmental studies/applications would help us evaluate the contribution of your dataset to the scientific community or beyond. Based on the current literatures in mapping urban land use and land cover change at annual interval, I think the presented dataset may be of limited use to characterize duration, change magnitude, and timing of urbanization.

Additionally, the methods presented in this manuscript are not in "best practices". For example, the reference impervious surface fractions used to build regression models in this study are extracted from spectral mixture analysis (obviously extracted manually). This seems to be against what the authors claimed in the Introduction section that manual extraction of endmembers may lead to biased estimations of ISA and vegetation fractions (it should have biased estimations). At least, I think the authors should provide an assessment of the reference ISA fractions (similar to what you did for final datasets) used to build the model at each city and how uncertainties/errors from this subjective reference dataset can eventually propagate to the final ISA and vegetation fraction dataset. Anyway, I think the authors should provide estimates of errors and

uncertainties associated with this dataset (which is related to data quality in question 2). It is worth noting that spectral mixture analysis is recently standardized at global scale and can be used to estimate ISA and vegetation fractions at an annual interval (e.g., Small 2013 in Remote Sensing of Environment).

I do not quite agree with the authors that the dataset provides metrics for urban structure. This is confusing since urban structure may more refer to its landscape patterns, where shopping malls are located and where residential areas are located. The dataset only refers to the landscape composition in urban areas.

2. Is the data set accessible via the given identifier? Is the data set complete? Are error estimates and sources of errors given (and discussed in the article)? Are the accuracy, calibration, processing, etc. state of the art? Are common standards used for comparison?

The dataset is accessible and complete as described in the manuscript. As the authors refined the existing dataset for generating ISA and vegetation fractions, the accuracy of the presented dataset should be also dependent on the accuracy of the previous dataset. Thus, the final reported accuracy should be the product of the accuracy of the previous dataset and the newly generated dataset. Further accuracy assessment of this dataset should be reported.

In comparison with other global urban datasets as shown in Fig. 8, I think the dataset from this manuscript is not as accurate as ESA land cover dataset. It seems that the new dataset sets a hard boundary for urban areas and discard neighboring regions beyond the boundary. This dataset is then may be of further limited use for studies in climate modeling (e.g., in WRF) that requires continuous land cover datasets in both spatial and temporal domains.

3. Are there any inconsistencies within these, implausible assertions or data, or noticeable problems which would suggest the data are erroneous (or worse). If possible, apply tests (e.g. statistics). Unusual formats or other circumstances which impede

such tests in your discipline may raise suspicion.

The spatial resolution is not consistent. The manuscript claimed it at 30 m, but what I see from the dataset is 250 m.

The only comparison I can think of, which the author can do, is a comparison between your dataset and other existing global dataset in terms of changes in urban areas over time (rather than just simple visual comparisons of maps). Specific numbers from each dataset for selected cities can help us further evaluate the performance of the method and the dataset. But this is a minor comment.

4. Is the data set usable in its current format and size? Are the formal metadata appropriate? Check the publication: Is the length of the article appropriate? Is the overall structure of the article well structured and clear? Is the language consistent and precise? Are mathematical formulae, symbols, abbreviations, and units correctly defined and used? Are figures and tables correct and of high quality?

I would suggest the authors add more metadata to describe the dataset in the downloaded documents: spatial resolution, extent, cities included, accuracy for each city, legend. The dataset I downloaded from the website does not include that information although brief information is available on the website.

Figure 4 can be improved. I do not really understand what Fig. 4 tells us: is the logistic regression is the right method to use? Maybe random forest regression is better?

I am not clear of what criteria you used to apply your built models to other cities. Based on locations? How practical for this method to be applied at broad scale or national scale? This approach can be easily improved with more automatic methods for example using globally standardized spectral mixture analysis (Small et al. 2013 in Remote Sensing of Environment. Thus, the method you used does not fit in the "uniqueness" point as identified on the publisher's website, see the reviewer guidelines).

---

## Referee Comment (RC2) · Anonymous Referee #2 · 4 Oct 2019

This study proposed a method to provide multi-year urban land cover maps for China. However, the proposed method suffers from several critical issues and uncertainties remain for the resulted maps. Thus, I do not recommend this manuscript to be accepted for publication in the journal.

First, my main concern is the accuracy of these maps to detect urban land cover changes. It is more important for readers to know the accuracy of change detection than merely mapping accuracy of each year because this is the reason why we need maps at multiple time points. Meanwhile, CLUD was created by visual interpretation and ignored small urban clusters and human settlements as indicated by Fig. 8. Thus, the maps generated in this study seem to loss the details that the 30-m resolution Landsat images can provide. In addition, the created maps should also be compared

to more existing urban maps with higher spatial resolution than the ones used in this study (Fig. 8), such as the Global Human Settlement Layer and Global Urban Footprint.

The method that used NDVI as the single indicator to estimate surface imperviousness is problematic due to the confusion between bare land and impervious surfaces. This is especially true for China where a large amount of bare land existed due to rapid urbanization during the study period. Although the authors used EBBI to extract bare land, the capability of this index to differentiate impervious surfaces and bare land across biomes is uncertain.

The method used to temporally adjust multi-year ISA is also questionable because urban redevelopment (e.g., convert high ISA urban villages to high residential buildings with vegetation) is very common in Chinses cities. Thus, the inversible assumption of urban development is problematic, especially for this study that aimed to detail intra-urban land cover change. Additionally, the authors mentioned that the 2015 map was the most accurate so that it was used as reference to temporally adjust ISA in previous years. Quantitative results were needed to support this decision and whether the 2015 map was the most accurate across regions should be addressed.

Line 2, Page 3: 30-m resolution is not "high-resolution" Line 15, page 5: not just "Javascript"

---

## Referee Comment (RC3) · Anonymous Referee #3 · 10 Oct 2019

1 Page 4 line 4-10: Please explain how the method used by CLUD to extract urban areasïïj§Machine learning or manual interpretation? How does the errors on the boundary affect the analysis results of the proposed data set?

2 Page 4 line 4-10: "With prior knowledge of image classification and humancomputer visual interpretation, we extracted urban land in Suzhou by detecting the city's boundaries", the authors used CLUD to extract the city boundarty, but the prior knowledge and visual interpretation were used to extract the boundary of Suzhou. Do all cities in the dataset used the boundaries with visual interpretationïïj§

3 Page 4 line 13: What's ISA fraction? The fraction should be appeared in the manuscript.

[Figure]

4 Page 4 line 24-25: "In addition, the input parameters required by logistic regression—ISA classification data and NDVI maximum data—can be obtained through existing methods and datasets", It would be better to to explain what parameters they are.

5 Page 5 line23-25: "The spectral unmixing method was employed to unmix the Landsat multispectral bands into the four endmembers. A decision tree was built to classify the high-albedo surfaces, low-albedo surfaces, water, vegetation and bare soil based on the fractions after unmixing and the calculation of indexes" , how did the authors use the MNF to process different remote sensing images which have different atmospheric conditions?

6 Page6 line 11-22: It is would be better to to use high resolution images to interpret the green space and then verify the accuracy of UGS.

7 Page6 line 29-30: "Because of its relatively high accuracy...", how high is the accuracy?

8 Page 8 line 11-19 : "In our dataset, the urban and rural areas are well distinguished because of a good definition of urban area", the authors should provide more convincing evidence to prove the good definition of urban area.

---

## Author Comment (AC1) · 8 Nov 2019

**General Comment:**

A 30-meter resolution national urban land-cover data-set of China, 2000–2015 will give opportunity to conduct interesting research related to the topic of urban climate and the contribution of the land use land cover change including historical urbanization in the climate evolution at regional scale.

**Response:**

Yes, this product will be valuable for many applications such as urban climate and environmental analysis. We added texts in the Introduction section to indicate potential applications.

---

## Author Comment (AC2) · 8 Nov 2019

**General Comment:**

Kuang et al. refined an existing land use and land cover data set (China's Land Use/cover Dataset) specifically for generating fractions of impervious surface area (ISA) and vegetation (within cities) at national level. While I find the manuscript and the dataset of general interest, I still have some concerns that I think the authors need further consideration. I will comment on this manuscript following the guidelines from the publisher's website.

**Response:**

Thank you very much for your constructive comments. We revised the manuscript according to your comments and suggestions. Detailed response and changes in

manuscript were listed in the PDF file in the supplement.

**Section 1**
**Section 1, Paragraph 1**
**Section 1, Paragraph 1, Point 1:** The data set related to ISA fraction and urban vegetation fraction at national level presented in the manuscript is new but not the method used to estimate them. The use of NDVI and other auxiliary data including reflectance to estimate ISA fraction has been done previously (e.g., Sexton et al., 2013, Remote Sensing of Environment). Obviously, these types of citations are neglected in the manuscript.

**Response:**
Thank you for your comments. We found that previous studies mainly focused on the analysis of urban land covers at individual city scale. For example, Sexton et al. used a single regression model to retrieve ISA in Washington, D.C.-Baltimore MD (Sexton et al., 2013, RSE). Our case focused on mapping of intra-urban land-cover at a national extent with the support of GEE, which is much complex. In this revised manuscript, we added more text to describe the methods and added more key references.

**Changes in manuscript:**
We revised the related sentences in the introduction part and cited more references, including Sexton et al. (2013) in page 3, lines 1-10.

**Section 1, Paragraph 1, Point 2:** Another factor that may lead to the judgement of the data set presented in this manuscript not as useful as it claimed by the authors is the mapping interval (5-year). A quick search of the current literatures would tell you that the scientific community now advocates for urban land cover datasets at a higher temporal frequency (e.g., annual mapping), particularly for urban environmental and climate studies. However, the authors did not even identify/mention possible use of

their datasets of a five-year interval. For example, how does your dataset contribute to "world urban database" that may eventually help studies in urban climate using earth system models (e.g., weather research and forecasting, WRF)? I am not advocating for a case study or specifically linking your dataset to "world urban database", but a potential linkage between this presented dataset and environmental studies/applications would help us evaluate the contribution of your dataset to the scientific community or beyond. Based on the current literatures in mapping urban land use and land cover change at annual interval, I think the presented dataset may be of limited use to characterize duration, change magnitude, and timing of urbanization.

**Response:**

Thanks for your comments. Yes, we agree entirely with you that annual datasets have higher values than five or ten-year datasets. Currently ESA- and MODIS-based annual land cover products and Landsat-based urban datasets were generated. However, the ESA- and MODIS-based datasets cannot effectively capture urban spatial patterns due to coarse spatial resolution, while Landsat-based urban datasets have relatively low accuracy (for example, producer's accuracy and user's accuracy are 0.50–0.60 and 0.49–0.61, Liu et al. 2018, RSE) that cannot meet requirements of real applications. In order to produce high-spatial resolution and high accurate urban datasets, we integrated different data sources and approaches to produce China's urban datasets that can meet real applications. Because of time-consuming and intensive labor, it is challenging to generate annual datasets. We think five-year urban dataset is suitable considering the following reasons: (1) urban expansion often occurred dispersedly and in relatively small patch sizes in a year, thus Landsat images with 30 m spatial resolution cannot effectively capture this kind of changes in annual time interval; (2) urban datasets are often related to socioeconomic data, which they are often surveyed at five or ten-year interval; (3) most of national land cover products such as US National Land Cover Database (NLCD) released at five-year or ten-year interval (1992, 2001, 2006, 2011, 2016) (Yang et al., 2018, ISPRS Journal of Photogrammetry and Remote Sensing) and China's Land Use/Cover Dataset (CLUD) at five-year interval (Liu et al.,

2005, RSE; Zhang et al., 2016, RSE), considering time and labor.

In the original manuscript, we did not discuss the use of this product. Thanks for your suggestion, we added texts to indicate the potential uses of this product in different fields such as environments, urban climate, human settlements management, socioeconomic analysis, and future planning of national urban development.

**Changes in manuscript:**

We revised the abstract of the manuscript. We also added texts in the introduction section to indicate the potential use of this product in different fields.

**Section 1, Paragraph 2**

**Section 1, Paragraph 2, Point 1:** Additionally, the methods presented in this manuscript are not in "best practices". For example, the reference impervious surface fractions used to build regression models in this study are extracted from spectral mixture analysis (obviously extracted manually). This seems to be against what the authors claimed in the Introduction section that manual extraction of endmembers may lead to biased estimations of ISA and vegetation fractions (it should have biased estimations). At least, I think the authors should provide an assessment of the reference ISA fractions (similar to what you did for final datasets) used to build the model at each city and how uncertainties/errors from this subjective reference dataset can eventually propagate to the final ISA and vegetation fraction dataset. Anyway, I think the authors should provide estimates of errors and uncertainties associated with this dataset (which is related to data quality in question 2).

**Response:**

Thank you for your comments and suggestions. The ISA dataset was generated using the same approach that was detailed in our previous publication (Kuang et al., 2014, Landscape and Urban Planning). The results were validated using reference data and an overall accuracy of 91.1% was obtained. More texts were added to indicate the accuracy issue.

**Changes in manuscript:**
We added the accuracy of ISA classification in the revised manuscript, see page 6, line 5.

**Section 1, Paragraph 2, Point 2:** It is worth noting that spectral mixture analysis is recently standardized at global scale and can be used to estimate ISA and vegetation fractions at an annual interval (e.g., Small 2013 in Remote Sensing of Environment).
**Response:**
Yes, spectral mixture analysis is a powerful tool for decomposing multispectral imagery into different fractional images. As Small indicated that globally standardized spectral mixture analysis can effectively extract substrate, dark and vegetation. However, ISA cannot be accurately and directly extracted from multispectral image using spectral mixture analysis considering the wide spectral variation of ISA, that is, similar spectral signatures between ISA and other non-vegetation types, such as bare soils and water. Also the meaning of substrate and dark used in Small (2013) is different with ISA.

**Section 1, Paragraph 3**
**Section 1, Paragraph 3, Point 1:** I do not quite agree with the authors that the dataset provides metrics for urban structure. This is confusing since urban structure may more refer to its landscape patterns, where shopping malls are located and where residential areas are located. The dataset only refers to the landscape composition in urban areas.
**Response:**
There is a little confusion about the urban structure in this manuscript. In the revised version, we replaced "urban structure" with "intra-urban land-cover". Thanks for your comments.
**Changes in manuscript:**
We revised the manuscript and replaced "urban structure" with "intra-urban land-cover",

which can be found in page 1 line 11, page 2 line 25, page 2 line 30, page 3 line 2, page 3 line 27, page 3 line 30, page 6 line 1, page 9 line 13, page 10 line 27, page 11 line 2.

**Section 2**
**Section 2, Paragraph 1**
**Section 2, Paragraph 1, Point 1:** The dataset is accessible and complete as described in the manuscript. As the authors refined the existing dataset for generating ISA and vegetation fractions, the accuracy of the presented dataset should be also dependent on the accuracy of the previous dataset. Thus, the final reported accuracy should be the product of the accuracy of the previous dataset and the newly generated dataset. Further accuracy assessment of this dataset should be reported.
**Response:**
Yes, we agree entirely with you that the accuracy in the previous dataset will affect the accuracy of the final results. Although the accuracy of previous dataset is high enough in the view of pixel level, the 30 m spatial resolution of pixel-level ISA data still contains a mixture of ISA and greenness (or even bare soil, water) because of the complex urban landscape. Therefore, we used the logistic regression approach to modify the pixel-level ISA data, then to produce fractional ISA dataset in order to improve the area statistics. We added the accuracy assessment results in the revised manuscript.
**Changes in manuscript:**
We added the accuracy assessment results of this dataset, see page 6, line 5.

**Section 2, Paragraph 2**
**Section 2, Paragraph 2, Point 1:** In comparison with other global urban datasets as shown in Fig. 8, I think the dataset from this manuscript is not as accurate as ESA land cover dataset. It seems that the new dataset sets a hard boundary for urban areas and

discard neighboring regions beyond the boundary. This dataset is then may be of further limited use for studies in climate modeling (e.g., in WRF) that requires continuous land cover datasets in both spatial and temporal domains.

**Response:**

Thank you for your comments. In our research, we focused on urban area and excluded the area without a sufficient population size. Therefore, we have the clear boundary of urban extent. For other global ISA datasets such as the ESA land cover dataset, they are valuable for global environmental studies, but these datasets have some shortcomings such as coarse spatial resolution resulting in poor spatial patterns of urban land covers (ISA, greenness, water) and relatively low accuracy in the urban landscape. Our objective is to provide accurate urban ISA and greenness datasets with much higher spatial resolution (30 m in our study). In order to compare different datasets, we summarize the current urban land products to delineate different among them (Tabel 1, shown in Fig. 1 below) and provides an example figure to show the area statistics based on Beijing city. Because other products can't effectively distinguish urban and rural lands, their urban areas were considerably overestimated (Figure 1, shown in Fig. 2 below). Based on accuracy assessment of our results, we obtained accuracy range between 92.0% and 98.9%, much higher accuracies than other existing products.

In addition, our dataset is developed from CLUD. The rural area is presented in CLUD (Figure 2, shown in Fig. 3 below). They can be integrated into our results if needed.

**Section 3**

**Section 3, Paragraph 1**

**Section 3, Paragraph 1, Point 1:** The spatial resolution is not consistent. The manuscript claimed it at 30 m, but what I see from the dataset is 250 m.

**Response:**

Yes, we developed the 30 m spatial resolution products, but the uploaded dataset was resampled to 250 m spatial resolution, considering the data size. The

30 m resolution dataset will be available by contacting the corresponding author
(kuangwh@igsnrr.ac.cn)

**Section 3, Paragraph 2**
**Section 3, Paragraph 2, Point 1:** The only comparison I can think of, which the author
can do, is a comparison between your dataset and other existing global dataset in
terms of changes in urban areas over time (rather than just simple visual comparisons
of maps). Specific numbers from each dataset for selected cities can help us further
evaluate the performance of the method and the dataset. But this is a minor comment.
**Response:**
Good suggestion, thanks. We conducted a comparison of different products based on
Beijing city, as replied in Section 2, Paragraph 2, Point 1.
**Changes in manuscript:**
We added the figure in section 4.2 of the manuscript. The manuscript was revised to
provide detailed explanation.

**Section 4**
**Section 4, Paragraph 1**
**Section 4, Paragraph 1, Point 1:** I would suggest the authors add more metadata to
describe the dataset in the downloaded documents: spatial resolution, extent, cities
included, accuracy for each city, legend. The dataset I downloaded from the website
does not include that information although brief information is available on the website.
**Response:**
Thanks for your suggestion. We revised the metadata of the website so that readers
can obtain detailed information about this product.
**Changes in manuscript:**
We resubmitted the metadata file on the website.

**Section 4, Paragraph 2**

**Section 4, Paragraph 2, Point 1:** Figure 4 can be improved. I do not really understand what Fig. 4 tells us: is the logistic regression is the right method to use? Maybe random forest regression is better?

**Response:**

Figure 4 showed the logistic regression model of impervious surface estimation based on four cities – Dalian, Jinan, Wuhan, and Xi'an. The left (orange) and middle (green) histograms showed the frequency distribution of NDVImax value for ISA and non-ISA sample points, respectively. The right figure (blue curve) showed the logistic regression model fitted with the sample points in the left and medium figures. For different regions or cities, the regression models vary. We built different models in China for ISA estimation. For example, the regression curve of Xi'an showed a steep slope and Dalian a relatively smooth slope.

Random Forest (RF) is a commonly used machine learning method for urban land classification and ISA estimation when multiple variables were used. However, when only one variable was used, RF does not have the advantage over other methods. In particular, when training samples are only located some specific regions, RF-based model cannot be effectively transferred to other regions without training samples. Considering that our study is to establish a model based on one variable and this model will be used to estimate ISA at national scale, we selected the logistic regression approach to estimate ISA in order to effectively use this model to estimate different cities. Based on our exploration in limited number of cities, the logistic regression model provided accurate estimation with RMSE of 0.1.

**Section 4, Paragraph 3**

**Section 4, Paragraph 3, Point 1:** I am not clear of what criteria you used to apply your built models to other cities. Based on locations? How practical for this method to be applied at broad scale or national scale?

**Response:**

Good question. In our original version, we did not clearly describe this issue. So in the revised one, we added more texts to explain this issue. In China, population density and economic condition have wide variation, resulting in considerably different ISA distribution across the country. In order to solve this problem, we developed different models according to specific economic regions. For example, we choose the Chinese economic geographic zones and assumed a consistent logistic regression relationship within each partition. A certain number of cities were selected and the logistic regression parameters of each city were calculated. The average value of the parameters in each economic and geographic zone is obtained as a regression parameter for all cities in the same zone. Based on this method, we calculated the preliminary ISA value for cities in each zone.

**Changes in manuscript:**

More texts were added to provide the explanation, see page 6, lines 7-18.

**Section 4, Paragraph 4, Point 2:** This approach can be easily improved with more automatic methods for example using globally standardized spectral mixture analysis (Small et al. 2013 in Remote Sensing of Environment. Thus, the method you used does not fit in the "uniqueness" point as identified on the publisher's website, see the reviewer guidelines).

**Response:**

As replied in Section 1, Paragraph 2, Point 2, the globally standardized spectral mixture analysis is a valuable tool to provide a standard method for land use classification, but it cannot effectively and directly extract ISA datasets without intensive post-processing. Therefore, we proposed the integrated approach to provide accurate ISA and greenness datasets, although this approach takes much time and labor to produce the product.

[Figure]

Please also note the supplement to this comment:
https://www.earth-syst-sci-data-discuss.net/essd-2019-65/essd-2019-65-AC2-supplement.pdf

[Figure]

| Name | Spatial resolution | Abbreviation | Method | Reference |
|---|---|---|---|---|
| China's Urban Land use/cover Dataset | 30 m | CLUD-Urban | Visual interpretation | - |
| Land Cover from Moderate-resolution Imaging Spectroradiometer | 500 m | MODIS LC | Decision tree classification | (Friedl et al., 2010) |
| European Space Agency global land-cover data | 300 m | ESA LC | Unsupervised classification and change detection | (Bontemps et al., 2011) |
| Global Land Cover at 30 m resolution | 30 m | GlobaLand 30 | Pixel-Object Knowledge (POK)-based classification | (Chen et al., 2015) |
| Built-up grid of the Global Human Settlement Layer | 30 m | GHS Built | Symbolic machine learning | (Pesaresi et al., 2013, 2016) |
| Multi-temporal Global Impervious Surface | 30 m | MGIS | Normalized urban areas composite index | (Liu et al., 2018) |

**Fig. 1.** Table 1: List of urban land products for comparison.

[Figure]

**Fig. 2.** Figure 1: Comparison of urban land area and change in Beijing city based on different urban land-use products.

China's Land use/cover Dataset (CLUD)

China's Urban Land use/cover Dataset (CLUD-Urban)

Beijing city, 2015

Beijing city, 2015

| | Cropland | | Grass | | Urban land | | Bare land |
| | Waterbody | | Forest | | Rural area and industrial land |

Urban ISA fraction (%)

0  10 20 30 40 50 60 70 80 90 100

0        10 km

**Fig. 3.** Figure 2: Comparison of former CLUD (left figure) and the newly developed CLUD-Urban (right figure, delineating detailed intra-urban land-cover).

---

## Author Comment (AC3) · 8 Nov 2019

**General Comment:**
This study proposed a method to provide multi-year urban land cover maps for China. However, the proposed method suffers from several critical issues and uncertainties remain for the resulted maps. Thus, I do not recommend this manuscript to be accepted for publication in the journal.

**Response:**
Thanks for your comments. In the original version, we did not clearly describe the methods and results; thus, the true values of these products cannot be effectively evaluated. In the revised manuscript, we provided more detailed explanation of the methods and results, including accuracy assessment. Thanks for your comments and suggestions,

as well as from other reviewers, we believe the new version is considerably improved. Detailed response and changes in manuscript were listed in the PDF file in the supplement.

**Paragraph 1**

**Paragraph 1, Point 1:** First, my main concern is the accuracy of these maps to detect urban land cover changes. It is more important for readers to know the accuracy of change detection than merely mapping accuracy of each year because this is the reason why we need maps at multiple time points.

**Response:**

Yes, accuracy of the maps is very important. We did not pay sufficient attention in the original manuscript. Thanks for your suggestions, we conducted accuracy assessment and added tables (Tables 2 and 3) to explain this issue. The accuracy of the urban land-cover changes has been validated in CLUD (Liu et al., 2005, RSE; Zhang et al., 2014, RSE; Ning et al., 2018, JGS). The accuracies of urban land change in 2000-2005, 2005-2010, 2010-2015 were 97.01%, 95.93% and 94.99%, respectively. The accuracy of urban land for each period has range between 91.9% and 98.9%. The intra-urban land-cover dataset is derived from CLUD. The validation results showed that the RMSE values of ISA for 2000, 2005, 2010, and 2015 were 0.12, 0.11, 0.10, and 0.10, respectively. The RMSE values of UGS for 2000, 2005, 2010, and 2015 were 0.18, 0.17, 0.16, and 0.14, respectively.

We also performed the validation for change detection results for different period products of ISA and UGS fraction. We chose 741 samples (90m×90m) within urban area for validation. We used medium relatively error (MRE) to examine the accuracy. The MRE for ISA in 2000-2005, 2005-2010, and 2010-2015 were 5.69%, 5.33%, and 6.83%, respectively. The MRE values for UGS in 2000-2005, 2005-2010, and 2010-2015 were 5.69%, 5.33%, and 6.83%, respectively.

**Changes in manuscript:**

The validation results of ISA and UGS changes were also added in the validation section (4.1.2) of the manuscript. We revised Table 3 in the manuscript (Table 1, shown in Fig.1 below) to add the accuracy of urban land-cover types for 2000, 2005, 2010 and 2015.

We added a new table in the revised manuscript to delineate the accuracy of urban land and its intra land-cover change (Table 2, shown in Fig.2 below).

**Paragraph 1, Point 2:** Meanwhile, CLUD was created by visual interpretation and ignored small urban clusters and human settlements as indicated by Fig. 8. Thus, the maps generated in this study seem to loss the details that the 30-m resolution Landsat images can provide.

**Response:**

Yes, we agree with you that the CLUD loss the details in the urban landscape, because one objective of producing CLUD was to obtain the urban extent, not spatial details in the urban landscape. However, the CLUD has accurate urban boundary through visual interpretation and this research just used this advantage. This research aims to making use of advantages of different data sources to produce accurate and detailed urban components. Therefore, we used CLUD to provide the accurate urban boundary and urban pixels, then used other data sources (such as NDVImax, fractional ISA reference data) to establish models to estimate fractional ISA, as well as green spaces, so that we can provide the detailed urban components, which other individual data sources do not have.

**Paragraph 1, Point 3:** In addition, the created maps should also be compared. to more existing urban maps with higher spatial resolution than the ones used in this study (Fig. 8), such as the Global Human Settlement Layer and Global Urban Footprint.

**Response:**

Good suggestion. Thanks a lot. We conducted the comparison of our results with existing urban land datasets, as shown in Fig. 8 (in the manuscript). A list of urban land datasets includes MODIS land cover, ESA land cover, Global Human Settlement Layer (GHSL), GlobaLand 30 and other products were compared (Table 3, shown in Fig.3 below). As shown in Figure 8 in the manuscript (Figure 1, shown in Fig.4 below), our product provided more details of urban spatial patterns than other products (note: both of the GHS Built and GlobaLand 30 products only have two years).

**Changes in manuscript:**
We revised Fig. 8 in the manuscript and added more urban land products at 30 m or higher spatial resolution. The manuscript was revised to add a more detailed comparison in section 4.2.

**Paragraph 2**
**Paragraph 2, Point 1:** The method that used NDVI as the single indicator to estimate surface imperviousness is problematic due to the confusion between bare land and impervious surfaces. This is especially true for China where a large amount of bare land existed due to rapid urbanization during the study period. Although the authors used EBBI to extract bare land, the capability of this index to differentiate impervious surfaces and bare land across biomes is uncertain.

**Response:**
We completely agree with you that bare land is a big problem in ISA mapping, especially in arid/semiarid regions. This is a common problem and there is lack of suitable approaches to solve this problem yet. This problem can be ignored in tropical and subtropical regions as we used NDVImax, but cannot be ignored in arid and semiarid regions. In this research, we explored use of EBBI bare land index to remove bare land (As-syakur et al., 2012, Remote Sens.). More research should be conducted in the future to explore suitable approach to separate bare land and ISA. If high spatial resolution thermal images or nighttime light data are available in the future, proper

integration of these types of data may be a solution. We added some texts in the Discussion section to discuss this issue.

**Changes in manuscript:**
We added texts in Discussion section (section 4.6) to discuss the methods or data which can be improved in further research.

**Paragraph 3**
**Paragraph 3, Point 1:** The method used to temporally adjust multi-year ISA is also questionable because urban redevelopment (e.g., convert high ISA urban villages to high residential buildings with vegetation) is very common in Chinses cities. Thus, the inversible assumption of urban development is problematic, especially for this study that aimed to detail intra-urban land cover change.

**Response:**
Good comments. We agreed with you on this concern. If we directly conduct the temporal adjustment, the redevelopment area has problem. However, we used NDVImax to do the calibration. In the redevelopment region, if ISA was replaced with greenness (such as park), this kind of problem is solved. When we conducted the calibration, we assume that the generally increasing trend of impervious density inside Chinese cities is mainly in the urban greening area. We modified the manuscript to stress this assumption.

**Changes in manuscript:**
We stressed that the temporal adjustment between multi-year ISA data was conducted. The related change can be found in page 6, line 33, page 7, line 2.

**Paragraph 3, Point 2:** Additionally, the authors mentioned that the 2015 map was the most accurate so that it was used as reference to temporally adjust ISA in previous years. Quantitative results were needed to support this decision and whether the 2015

map was the most accurate across regions should be addressed.

**Response:**
We added the accuracy assessment results in the revised manuscript. The RMSE values for 2000, 2005, 2010, and 2015 were 0.12, 0.11, 0.11, and 0.10, respectively. The dataset in 2015 has the lowest error.

**Changes in manuscript:**
In page 7 lines 3-4, we added the RMSE values for 2000, 2005, 2010, and 2015 in the revised manuscript.

**Paragraph 4**
**Paragraph 4, Point 1:** Line 2, Page 3: 30-m resolution is not "high-resolution"
**Response:**
We replaced"high-resolution" as "30-m resolution".
**Changes in manuscript:**
We replaced "high-resolution" as "30-m resolution" in page 3, line 12.

**Paragraph 4, Point 2:** Line 15, page 5: not just"JavaScript".
**Response:**
As Python can be used in GEE, it could be better not to emphasize a particular language. We revised the manuscript.
**Changes in manuscript:**
We deleted "JavaScript" in this sentence.

Please also note the supplement to this comment:
https://www.earth-syst-sci-data-discuss.net/essd-2019-65/essd-2019-65-AC3-supplement.pdf

[Figure]

| Year | Urban land | ISA | | UGS | | Water body | Bare land |
|---|---|---|---|---|---|---|---|
| | Overall accuracy | RMSE | R | RMSE | R | Overall accuracy | Overall accuracy |
| 2000 | 98.92% | 0.12 | 0.89 | 0.17 | 0.85 | 95.97% | 73.96% |
| 2005 | 97.01% | 0.11 | 0.89 | 0.17 | 0.87 | 96.98% | 73.72% |
| 2010 | 93.99% | 0.1 | 0.91 | 0.16 | 0.87 | 98.66% | 76.19% |
| 2015 | 91.98% | 0.09 | 0.93 | 0.02 | 0.89 | 98.32% | 70.01% |

Note: The validations of urban land were obtained from Zhang et al.(2014), Kuang et al. (2016) and Zhang et al.(2019)

**Fig. 1.** Table 1: Accuracy assessments of CLUD-Urban.

| Period | Overall accuracy of Urban land | MRE of ISA | MRE of UGS |
|--------|-------------------------------|------------|------------|
| 2000-2005 | 97.01% | 5.69% | 7.09% |
| 2005-2010 | 95.93% | 5.33% | 5.86% |
| 2010-2015 | 94.99% | 6.83% | 6.68% |

Note: The validations of urban land were obtained from Zhang et al.(2014) and Ning et al.(2019)

**Fig. 2.** Table 2: Accuracy assessments of urban land-cover change.

| Name | Spatial resolution | Abbreviation | Method | Reference |
|------|--------------------|--------------|--------|-----------|
| China's Urban Land use/cover Dataset | 30 m | CLUD-Urban | Visual interpretation | - |
| Land Cover from Moderate-resolution Imaging Spectroradiometer | 500 m | MODIS LC | Decision tree classification | (Friedl et al., 2010) |
| European Space Agency global land-cover data | 300 m | ESA LC | Unsupervised classification and change detection | (Bontemps et al., 2011) |
| Global Land Cover at 30 m resolution | 30 m | GlobaLand 30 | Pixel-Object Knowledge (POK)-based classification | (Chen et al., 2015) |
| Built-up grid of the Global Human Settlement Layer | 30 m | GHS Built | Symbolic machine learning | (Pesaresi et al., 2013, 2016) |
| Multi-temporal Global Impervious Surface | 30 m | MGIS | Normalized urban areas composite index | (Liu et al., 2018) |

**Fig. 3.** Table 3: List of urban land products for comparison.

a) Landsat image   b) This dataset   c) MODIS LC   d) ESA LC   e) GlobaLand 30   f) GHS Built   g) MGIS

2000

2005

2010

2015

Cropland   Grass   Urban land or built-up area    Urban ISA fraction (%)        0    20 km

Waterbody   Forest   Rural area and industrial land    0  10 20 30 40 50 60 70 80 90 100

**Fig. 4.** Figure 1: Comparison of urban land distribution in Beijing, China, from different urban products.

---

## Author Comment (AC4) · 8 Nov 2019

**Point 1:** Page 4 line 4-10: Please explain how the method used by CLUD to extract urban area by Machine learning or manual interpretation? How does the errors on the boundary affect the analysis results of the proposed data set?
**Response:**
Our previous publications have detailed the approach for developing CLUD (Liu, Liu, Tian et al., 2005, RSE; Liu et al., 2009, JGS; Liu et al., 2014; JGS): This approach can be briefly summarized as following: (1) build the interpretation symbols of cities in Landsat images, (2) the polygons in GIS were used to depict urban boundaries. When human-computer visual interpretation was conducted, polygons were created and labelled as urban area, (3) the urban boundaries were revised according to urban

features like concentrated contiguously.

The accuracy of the proposed dataset includes two parts: the accuracy of urban boundaries and the accuracy of urban land-cover classification. The average accuracy of urban land was 95.48 %. The accuracy of urban boundaries reflects the accuracy of urban land extraction. As the manual interpretation was adopted, the urban boundaries are more accurate than others (Liu, Liu, Tian et al., 2005; Liu et al., 2009, JGS, Liu et al., 2014; JGS).

**Changes in manuscript:**

We added texts to briefly explain the method to produce CLUD, see page 4, lines 22-25.

**Point 2:** Page 4 line 4-10: "With prior knowledge of image classification and human computer visual interpretation, we extracted urban land in Suzhou by detecting the city's boundaries", the authors used CLUD to extract the city boundary, but the prior knowledge and visual interpretation were used to extract the boundary of Suzhou. Do all cities in the dataset used the boundaries with visual interpretation?

**Response:**

Yes, we used visual interpretation to identify all city boundaries. We took Suzhou city as an example to show how the method worked. We revised the manuscript to make the expression clearer.

**Changes in manuscript:**

In Page 4 line 21, we replaced "urban land in Suzhou" to "China's urban land".

**Point 3:** Page 4 line 13: What's ISA fraction? The fraction should be appeared in the manuscript.

**Response:**

ISA fraction refers to the percentage of ISA in a pixel. We added texts to explain ISA

fraction.

**Changes in manuscript:**
In Page 4 lines 28-29, we added a sentence to explain ISA and UGS fraction.

**Point 4:** Page 4 line 24-25: "In addition, the input parameters required by logistic regression via ISA classification data and NDVI maximum data can be obtained through existing methods and datasets", It would be better to explain what parameters they are.
**Response:**
ISA classification data refers to ISA in land-cover classification data. NDVI maximum data refers to NDVI composite data which is the maximum NDVI value from all Landsat images in a year. We added texts to explain this term.
**Changes in manuscript:**
In page 5 lines 9-11, we added texts to explain ISA classification data and NDVI maximum data.

**Point 5:** Page 5 line23-25: "The spectral unmixing method was employed to unmix the Landsat multispectral bands into the four endmembers. A decision tree was built to classify the high-albedo surfaces, low-albedo surfaces, water, vegetation and bare soil based on the fractions after unmixing and the calculation of indexes", how did the authors use the MNF to process different remote sensing images which have different atmospheric conditions?
**Response:**
The atmospheric correction using FLAASH was done before MNF. In reality, MNF was used to support the endmember selection for each image separately, so atmospheric condition for individual images will not affect the selection of endmembers.
**Changes in manuscript:**
In page 5 lines 29-30, we added some texts to describe the image preprocessing (atmospheric calibration) in the revised manuscript.

**Point 6:** Page 6 line 11-22: It is would be better to use high resolution images to interpret the green space and then verify the accuracy of UGS.
**Response:**
Yes, we interpreted green space from high spatial resolution images for some typical areas, and then used these data as a reference to evaluate the UGS from Landsat.

**Point 7:** Page 6, line 29-30: "Because of its relatively high accuracy...", how high is the accuracy?
**Response:**
The accuracy assessment was conducted for each year. The RMSE values for 2000, 2005, 2010, and 2015 results are 0.12, 0.11, 0.11, and 0.10, respectively. We revised the relevant part by adding the exact RMSE values.
**Changes in manuscript:**
In page 5, lines 3-4, we revised the manuscript by adding RMSE values for 2000, 2005, 2010, and 2015 results.

**Point 8:** Page 8 line 11-19: "In our dataset, the urban and rural areas are well distinguished because of a good definition of urban area", the authors should provide more convincing evidence to prove the good definition of urban area.
**Response:**
In our study, urban area is characterized as land for residential, commercial, industrial, recreational, and transportation in cities and towns. Rural area refers to other human settlement in countryside (Liu, Liu, Tian et al., 2005, RSE; Kuang et al., 2016, Landscape and Urban Planning). They showed different features in remote sensing images, as shown in Figure 1. The urban land-cover area is more complex than rural area (Figure 1, shown below).
**Changes in manuscript:**

We revised the manuscript in page 8, lines 26-29.

Please also note the supplement to this comment:
https://www.earth-syst-sci-data-discuss.net/essd-2019-65/essd-2019-65-AC4-
supplement.pdf
* * *
[Figure]

Central Point: 116°19'E, 39°54'N    2014/05/27        Central Point: 116°13'E, 40°18'N    2014/05/27

**Fig. 1.** Figure 1: Comparison of urban and rural areas in remote sensing images.